# Longitudinal plasma proteomics reveals biomarkers of alveolar-capillary barrier disruption in critically ill COVID-19 patients

Erik Duijvelaar ⓘ[1] ✉, Jack Gisby ⓘ[2], James E. Peters ⓘ[2], Harm Jan Bogaard ⓘ[1] & Jurjan Aman ⓘ[1] ✉

The pathobiology of respiratory failure in COVID-19 consists of a complex interplay between viral cytopathic effects and a dysregulated host immune response. In critically ill patients, imatinib treatment demonstrated potential for reducing invasive ventilation duration and mortality. Here, we perform longitudinal profiling of 6385 plasma proteins in 318 hospitalised patients to investigate the biological processes involved in critical COVID-19, and assess the effects of imatinib treatment. Nine proteins measured at hospital admission accurately predict critical illness development. Next to dysregulation of inflammation, critical illness is characterised by pathways involving cellular adhesion, extracellular matrix turnover and tissue remodelling. Imatinib treatment attenuates protein perturbations associated with inflammation and extracellular matrix turnover. These proteomic alterations are contextualised using external pulmonary RNA-sequencing data of deceased COVID-19 patients and imatinib-treated Syrian hamsters. Together, we show that alveolar capillary barrier disruption in critical COVID-19 is reflected in the plasma proteome, and is attenuated with imatinib treatment. This study comprises a secondary analysis of both clinical data and plasma samples derived from a clinical trial that was registered with the EU Clinical Trials Register (EudraCT 2020–001236–10, https://www.clinicaltrialsregister.eu/ctr-search/trial/2020-001236-10/NL) and Netherlands Trial Register (NL8491, https://www.trialregister.nl/trial/8491).

The pathophysiology of coronavirus disease 2019 (COVID-19) consists of a complex interplay of direct viral cytopathic effects and an aberrant host immune response leading to alveolar damage, endothelial dysfunction and organ injury[1,2]. Critical illness is characterised by local and systemic inflammatory responses which induce excess vascular permeability and thrombi deposition in the pulmonary vasculature, resembling acute respiratory distress syndrome (ARDS)[3]. In the CounterCOVID trial, imatinib did not reduce the time required for liberation from ventilation and supplemental oxygen in hospitalised patients with COVID-19[4]. However, a signal of clinical benefit was found in severely ill patients as evidenced by reductions in the length of ICU stay, duration of invasive ventilation and mortality[4]. Long-term follow-up data confirmed that imatinib significantly improved oxygen uptake and long-term survival[5]. The beneficial effects of imatinib in critically ill COVID-19 patients could be mediated through the modulation of innate immune responses and reversal of endothelial dysfunction[6]. However, the mechanisms underlying critical illness and the effects of imatinib require further investigation. Here, we investigate the

[1]Department of Pulmonary Medicine, Amsterdam University Medical Centers, Amsterdam Cardiovascular Sciences, Amsterdam, The Netherlands.
[2]Department of Immunology and Inflammation, Centre for Inflammatory Disease, Imperial College London, London, UK.
✉ e-mail: e.duijvelaar@amsterdamumc.nl; j.aman@amsterdamumc.nl

differential effects of critical COVID-19 on the plasma proteome and assess the effects of imatinib on the perturbations of proteins associated with critical illness.

Wide-angle characterisation of intricate biological processes can be achieved through large-scale proteomics[7]. Proteins are the functional molecules of cells and therefore constitute the target of most drugs. Proteomics can help improve our understanding of COVID-19, evaluate treatment effects, and be used to characterise clinical subphenotypes[6,8]. Several dysregulated biological processes, including myeloid dysfunction, inadequate or exhausted T-cell responses, neutrophil activation, deficient interferon signalling, macrophage dysregulation, and complement activation have been linked to the severity of COVID-19 illness in previous proteomic studies[9–13]. Although previous studies have provided a consistent description of the host immune response, the plasma proteomic profile of alveolar-capillary injury—as central features of critical COVID-19 illness—remains lacking. This may result from either a small sample size, small proportion of critically ill patients, the use of small or selected protein panels or the lack of longitudinal measurements. In addition, the causality of the pathways involved remains unclear as these previous studies have not evaluated the effect of therapy on the plasma proteome in a randomised setting. Identification of pathways that contribute to critical illness may highlight biomarkers that could be used for risk stratification and provide targets for the treatment of respiratory failure in COVID-19 and acute respiratory distress syndrome (ARDS).

In this study, we profiled the plasma proteome of 318 patients who were included in the CounterCOVID study with moderate COVID-19 severity at hospital admission and were prospectively followed for the development of critical illness. The protein abundance of 6385 unique proteins or protein complexes was assessed using the aptamer-based Somascan platform. We investigated the mechanisms underlying critical illness and explored whether the proteome complements clinical features in risk stratification at hospital admission. In addition, we assessed the effects of imatinib treatment and determined how these effects relate to the perturbations in critical illness. In short, critical illness was reflected in the plasma proteome by perturbations in pathways involved in the acute phase response, cellular adhesion, extracellular matrix turnover and tissue remodelling. Part of these perturbations were attenuated by imatinib treatment.

## Results

Of the 385 patients hospitalised between March 31, 2020, and January 4, 2021, 332 had at least one plasma sample available for proteomic analysis (Fig. S1). Quality control (Figs. S1–4, Tables S2,3) resulted in the removal of 55 samples that had aberrant protein abundance levels, mainly due to low volume or clogging. Samples from 162 patients randomly assigned to receive imatinib and 156 assigned to placebo were available for proteomic analysis (Figs. 1, S1). Samples were collected at hospital admission (baseline) and 3 days later (follow-up). Patients included in this study had similar demographic and clinical characteristics to patients who did not have a plasma sample available for proteomic analysis (Table S1). Clinical outcome data were available for every patient until 90 days after randomisation. As expected, given the randomised study design, the clinical status for patients was similar between the two treatment groups at baseline. At time of baseline blood collection, 293 (92%) patients were admitted to the ward and received supplemental oxygen through a nasal cannula or mask, 16 (5%) patients received high-flow nasal oxygen, one (0.3%) patient was invasively ventilated and 8 (3%) patients were hospitalised but did not receive oxygen supplementation (Table 1). During their disease course, 44 (14%) patients required invasive ventilation and 40 (13%) had a fatal outcome, totalling 69 (22%) patients who were considered to have critical COVID-19 (fatal outcome or invasive ventilation; Fig. 1). The groups of patients who developed critical illness contained a higher

proportion of smokers and diabetics, were older and had higher CRP, Hs-cTnT, NT-proBNP, and LDH levels, but lower eGFR, thrombocyte and lymphocyte counts at baseline (Table 1).

### The effect of critical illness on the plasma proteome

We first assessed whether the development of critical COVID-19 in the 28-day study period was reflected by changes in the plasma proteome. This was achieved by analysis of differential plasma protein abundance at baseline and follow-up. At baseline, 62 proteins had lower abundance in patients who developed critical illness, while 474 proteins had higher abundance (Fig. 2a, Supplementary data 2, Supplementary Fig. S5). The number of proteins with differential abundance in patients who developed critical illness was substantially higher at follow-up (2456) compared to baseline (536). At follow-up, 1300 proteins exhibited increased abundance, while 1156 proteins displayed decreased abundance in critical illness (Fig. 2b, Supplementary data 3). Among the 50 most significant proteins at follow-up were bone morphogenetic protein 10 (BMP10), interleukin-6 (IL-6), atrial natriuretic peptide (ANP), IL-18, fibroblast growth factor (FGF) 4, FGF11, endothelin-1, C-X-C motif chemokine ligand 10 (CXCL10), C-C motif chemokine 5 (CCL5), transforming growth factor beta 3 (TGF-β3), endothelin-1 and histones H2A and H2B (Fig. S6). The large majority (474 of 536 proteins) of the proteins dysregulated at baseline were also found to be differentially abundant at follow-up (Fig. 2c). At follow-up, a large proportion of patients already met the criteria for critical illness, and from only three patients a sample was unavailable because of death (Fig. 2d). Intubation had already occurred in 32 of 44 (73%) patients who were eventually intubated (Fig. 2d). Therefore, the higher number of proteins with differential abundance at follow-up likely reflects clinical deterioration. The two proteins with the largest fold change at baseline were CXCL10 and advanced glycosylation end product-specific receptor (AGER; Fig. 2e). At follow-up, CXCL10 and histone H2B type 3-B had the largest fold changes (Fig. 2e).

We next used pathfindR to identify pathways that underlie critical illness. Enriched terms at baseline included transcription, cell cycle, platelet activation and aggregation, tumour necrosis factor (TNF), TGF-β, vascular endothelial growth factor (VEGF) and platelet-derived growth factor receptor (PDGFR) signalling (Fig. S7, Supplementary data 9). Most of the pathway terms enriched at baseline, were also enriched at follow-up (265/289). Pathway enrichment of the differentially abundant proteins at follow-up revealed dysregulation of, amongst others, transcription, post-translational modification, cellular senescence, protein turnover, cell cycle pathways, and TGF-β, PDGFR and fibroblast growth factor (FGF) signalling pathways (Fig. 2f, Supplementary data 10). Proteins involved in cellular adhesion (integrin alpha-5, intercellular adhesion molecule 2 (ICAM-2) and fibronectin-1) and extracellular matrix organisation (nidogen-1, laminin-2 and integrin binding sialoprotein (IBSP)) were also dysregulated at day 3, albeit that their pathway terms were not amongst the 30 most significant (Fig. 2g, Supplementary data 9). Taken together, this data shows that clinical deterioration is reflected by temporal changes in the plasma proteome, with changes being more pronounced at follow-up.

### Proteins display early severity-dependent trajectories

The differences observed in the proteome between samples taken at baseline and follow-up reflect the emergence of biological processes in the early stages after hospitalisation that relate to critical illness. To identify proteins with different temporal trajectories in critical versus non-critical patients, we applied linear mixed models with an interaction term between time and critical illness. This approach revealed that the abundance of 334 proteins increased over time in patients with a critical illness relative to those with a non-critical course, whereas the abundance of 151 proteins decreased over time (Fig. S8, Supplementary data 4). Proteins with the greatest effect size or highest significance level included acute phase proteins, complement factors,

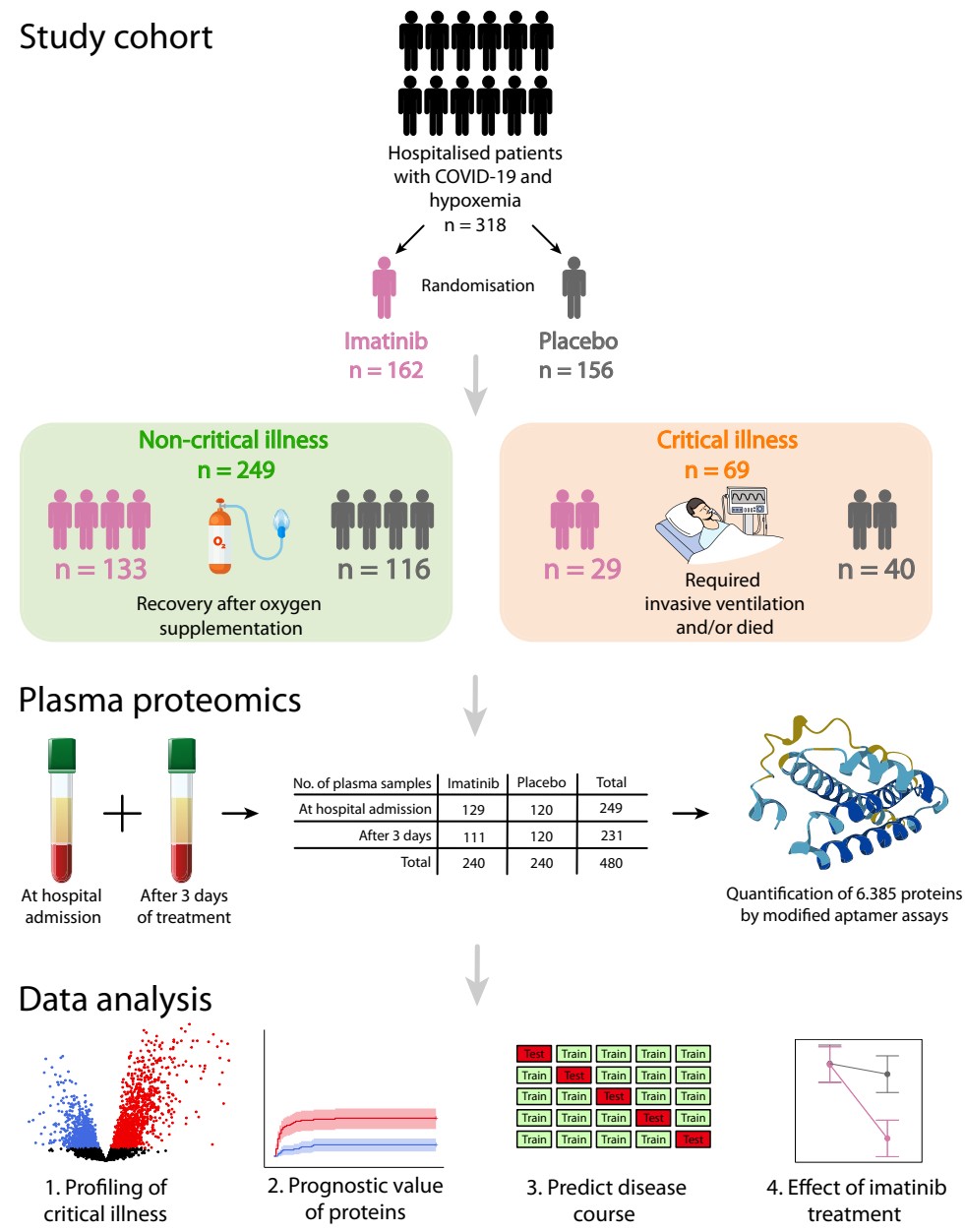

**Fig. 1 | Schematic overview of the study design.** Proteomic analyses were performed on plasma samples from a randomised double-blind placebo-controlled clinical trial. Patients who lacked a plasma sample were excluded from this study. Patients were divided into two groups: those with non-critical disease (i.e. those who recovered after receiving oxygen supplementation) and those with critical illness (i.e. those who required invasive ventilation and/or died). The total number of samples gathered over time is given.

chemokines, members of the TNF, FGF and TGF-β superfamilies, and proteins involved in cell-cell and cell-matrix adhesion and extracellular matrix (ECM) organisation (Fig. 3a). These proteins were enriched for pathways related to cell adhesion, extracellular matrix organisation, the complement cascade, and chemokine, TGF-β and TNF signalling (Fig. 3b, c, Supplementary data 11). The second most significant pathway was "SARS-CoV-2 Innate immunity evasion mechanisms and cell-specific immune responses; Metabolic overview over alveolar epithelial type II cells infection" (Fig. 3b)[14]. The above findings indicate that in critical illness, during the first days of hospital admission, there are dysregulations of ECM turnover, cell adhesion, remodelling and inflammatory cell activation and infiltration. These processes play key roles in inflammation, vascular leak and immunothrombosis in patients with COVID-19 related acute respiratory distress syndrome (ARDS)[15,16].

**Proteins at baseline associate with time to intubation or death**
We next sought to identify associations between protein abundance at baseline (*n* = 249) and time to development of critical illness. To this end, we used Cox proportional hazards regression and identified 538 proteins that were associated with time to critical illness (the 50 most statistically significant proteins are shown in Fig. 4a). For 467 proteins, abundance was positively associated with the risk for critical illness development, whereas 71 proteins were negatively associated (Fig. S9, Supplementary data 5). The proteins most strongly associated with risk of critical disease development were CXCL10 and AGER (Fig. 4b). Cadherin-3 (CDH3) and delta-like protein 3 (DLL3) were most negatively associated (Fig. 4b). Proteins that were significantly associated with critical illness development were enriched for pathways regulating mRNA processing and transcription, cell cycle, apoptosis, platelet activation and aggregation, T-lymphocyte activation as well as PDGF

## Table 1 | Patients stratified by critical disease development

| | Non-critical | Critical | p value |
|---|---|---|---|
| Patients, n | 249 | 69 | |
| **Demographic variables** | | | |
| Age, mean (SD) | 63.0 (12.1) | 69.2 (12.0) | 2.41e-04 |
| Male sex, n (%) | 167 (67.1) | 55 (79.7) | 0.061 |
| BMI, mean (SD) | 29.0 (5.4) | 29.8 (4.7) | 0.271 |
| History of smoking or current smoker, n (%) | 89 (37.1) | 35 (52.2) | 0.036 |
| Comorbid diabetes mellitus, n (%) | 58 (23.3) | 31 (44.9) | 0.001 |
| Comorbid cardiovascular disease, n (%) | 50 (20.1) | 20 (29.0) | 0.157 |
| Comorbid atrial fibrillation, n (%) | 18 (7.2) | 12 (17.4) | 0.020 |
| Comorbid hypertension, n (%) | 90 (36.1) | 30 (43.5) | 0.331 |
| **Treatment, n (%)** | | | |
| Imatinib | 133 (53.4) | 29 (42.0) | 0.124 |
| Dexamethasone | 177 (71.1) | 50 (72.5) | 0.941 |
| Remdesivir | 53 (21.3) | 9 (13.0) | 0.175 |
| Chloroquine | 23 (9.2) | 9 (13.0) | 0.481 |
| **Laboratory findings at admission** | | | |
| Haemoglobin (mmol/L), mean (SD) | 8.5 (0.9) | 8.4 (1.0) | 0.437 |
| Leucocytes (×10^9/L), mean (SD) | 8.5 (4.8) | 9.1 (4.5) | 0.365 |
| Neutrophils (×10^9/L), mean (SD) | 6.6 (3.3) | 7.5 (4.3) | 0.065 |
| Lymphocytes (×10^9/L), mean (SD) | 1.0 (0.5) | 0.9 (0.5) | 0.041 |
| Thrombocytes (×10^9/L), mean (SD) | 273 (110) | 229 (90) | 0.003 |
| eGFR (ml/min/1.73 m$^2$), mean (SD) | 79.3 (16.7) | 72.5 (20.7) | 0.005 |
| CRP (mg/L), mean (SD) | 102 (69) | 142 (90) | 0.001 |
| Hs-cTnT (ng/L), median [IQR] | 8.0 [4.0, 12.0] | 16.0 [8.0, 22.0] | 3.18e-05 |
| NTproBNP (ng/L), median [IQR] | 126 [37, 336] | 325 [88, 1075] | 0.006 |
| LDH (U/L), mean (SD) | 368 (140) | 471 (150) | 3.82e-06 |
| Albumin (g/L), mean (SD) | 35.2 (5.8) | 35.1 (5.3) | 0.875 |
| **Clinical characteristics at admission** | | | |
| SpO2 (%), mean (SD) | 94.3 (2.4) | 93.0 (2.9) | 0.001 |
| FiO2, mean (SD) | 32.4 (15.7) | 56.5 (24.2) | 1.26e-11 |
| SpO2/FiO2, mean (SD) | 328 (84) | 203 (95) | 2.2e-16 |
| modified WHO ordinal scale, n (%) | | | 0.001 |
| Hospitalised, no oxygen supplementation | 0 (0) | 8 (3.2) | |
| Oxygen mask or nasal cannula | 234 (94.0) | 59 (85.5) | |
| NIV or HFNO | 7 (2.8) | 9 (13.0) | |
| Invasive ventilation and additional organ support | 0 (0) | 1 (1.4) | |
| **Clinical course** | | | |
| length of admission (days), median [IQR] | 6.0 [3.0, 9.0] | 16.0 [7.0, 27.0] | 4.56e-12 |
| length of invasive ventilation (days), median [IQR] | 0 [0, 0] | 10.0 [4.8, 20.0] | 2.2e-16 |
| Need for ICU admission, n (%) | 13 (5.2) | 46 (66.7) | 2.2e-16 |
| Need for invasive ventilation, n (%) | 0 (0.0) | 44 (63.8) | 2.2e-16 |
| Mortality, n (%) | 0 (0.0) | 40 (58.0) | 2.2e-16 |

For normally distributed numeric variables (presented as mean (SD)), two-sided unpaired t-tests were used; for non-normally distributed numeric variables (median [IQR]), two-sided Mann-Whitney U tests were applied. Chi-square tests were used for categorical variables (n (%)). BMI body mass index, CRP C-reactive protein, eGFR estimated glomerular filtration rate, FiO2 fraction of inspired oxygen, HFNO high flow nasal oxygen, ICU intensive care unit, IQR interquartile range, LDH Lactate dehydrogenase, NIV non-invasive ventilation, SD standard deviation, SpO2 Peripheral oxygen saturation, WHO World Health Organization.

and TGF-β signalling (Fig. 4c, Supplementary data 12). Altogether, critical illness development is preceded by a dysregulation in proteins involved in the acute phase response, cell damage and tissue remodelling, which can already be detected at hospital admission.

## Using the baseline proteome to predict critical illness development

Next, we sought to identify a small set of proteins that could be measured at bedside for risk stratification. To this end, we used Least Absolute Shrinkage and Selection Operator (LASSO) regression to provide a sparse protein signature that could accurately predict the development of critical illness using protein abundance levels at baseline (n = 249). We trained the LASSO model by cross-validation in the training cohort, which consisted of all the participants except those that were patients at the Amsterdam UMC, location VUmc hospital (VUMC). These participants were held out to be used as an independent validation cohort (Fig. 4d). Patients from the validation cohort generally had a more severe disease course, which was inherent to our choice to select a subgroup of patients with a sufficient number of patients with a critical illness for appropriate validation. At baseline, CRP, LDH and NT-proBNP levels were higher in the validation group (Table S4). Moreover, patients from the validation cohort often had a more severe disease trajectory with a higher invasive ventilation incidence and longer hospital stays (Table S4).

During cross-validation in the training cohort, the model was able to predict the development of critical illness based on protein abundance levels with an estimated area under the curve (AUC) of 0.880 ± 0.04 (Fig. 4e). We additionally trained LASSO models based on clinical data (a list of included variables is provided in the data dictionary belonging to the Source data file) and a combination of proteomic and clinical data. The nine proteins that were selected by the final protein-only model were RNA binding protein fox-1 homologue 2 (RBFOX2), splicing factor 45 (RBM17), protein Dr1 (DR1), Prefoldin subunit 4 (PFDN4), translocon-associated protein subunit beta (SSR2), ataxin-2-binding protein 1 (RBFOX1), interleukin-6 (IL6), matrix extracellular phosphoglycoprotein (MEPE) and atrial natriuretic factor (NPPA). In the clinical-only model, the predictors included: thrombocyte count, oxygen saturation measured by pulse oximetry/fraction of inspired oxygen (SpO2/FiO2), comorbid diabetes mellitus, high-sensitive cardiac troponin T (hs-cTnT), N-terminal pro–B-type natriuretic peptide (NT-proBNP), age and SpO2. In the combined protein and clinical model, the selected predictors consisted of: SpO2/FiO2, DR1, RBFOX1 and PFDN4 (Fig. 4e). In the training cohorts, we observed higher performance of both the model based on the clinical data only (AUC = 0.919 ± 0.04) and that based on the combined protein and clinical data (AUC = 0.920 ± 0.04) (Fig. 4e). The predictors identified by each model are presented in Fig. 4e. When tested on the validation cohort, the AUC for the protein-only model was 0.813 (Fig. 4e). In the validation cohort, the protein-only model was superior to both the model based on clinical data (AUC = 0.688) and the model combining protein and clinical data (AUC = 0.712) (Fig. 4e). These findings indicate that, at hospital admission, the abundance of these nine proteins can be used for an accurate prediction of critical illness development, with superior predictive performance to the models based on clinical data in an independent cohort.

Subsequently, the prediction model underwent external validation across two independent study cohorts that employed the Somascan platform (Fig. S10)[17,18]. Due to the limited availability of clinical data from both study cohorts, the external validation relied exclusively on protein data. The first external validation cohort[18], encompassing 54 patients with end-stage kidney disease from the Hammersmith Hospital (HH), the model exhibited a comparable accuracy in predicting critical illness development (AUC = 0.801). The second external validation cohort[17] enrolled 308 patients from the Massachusetts General Hospital (MGH) who were hospitalised due to COVID-19. An earlier version of the Somascan assay was employed in the MGH cohort, therefore only five out of nine proteins from our signature set were available. For this reason, we assessed the performance of the 5-protein model in both our internal VUMC validation

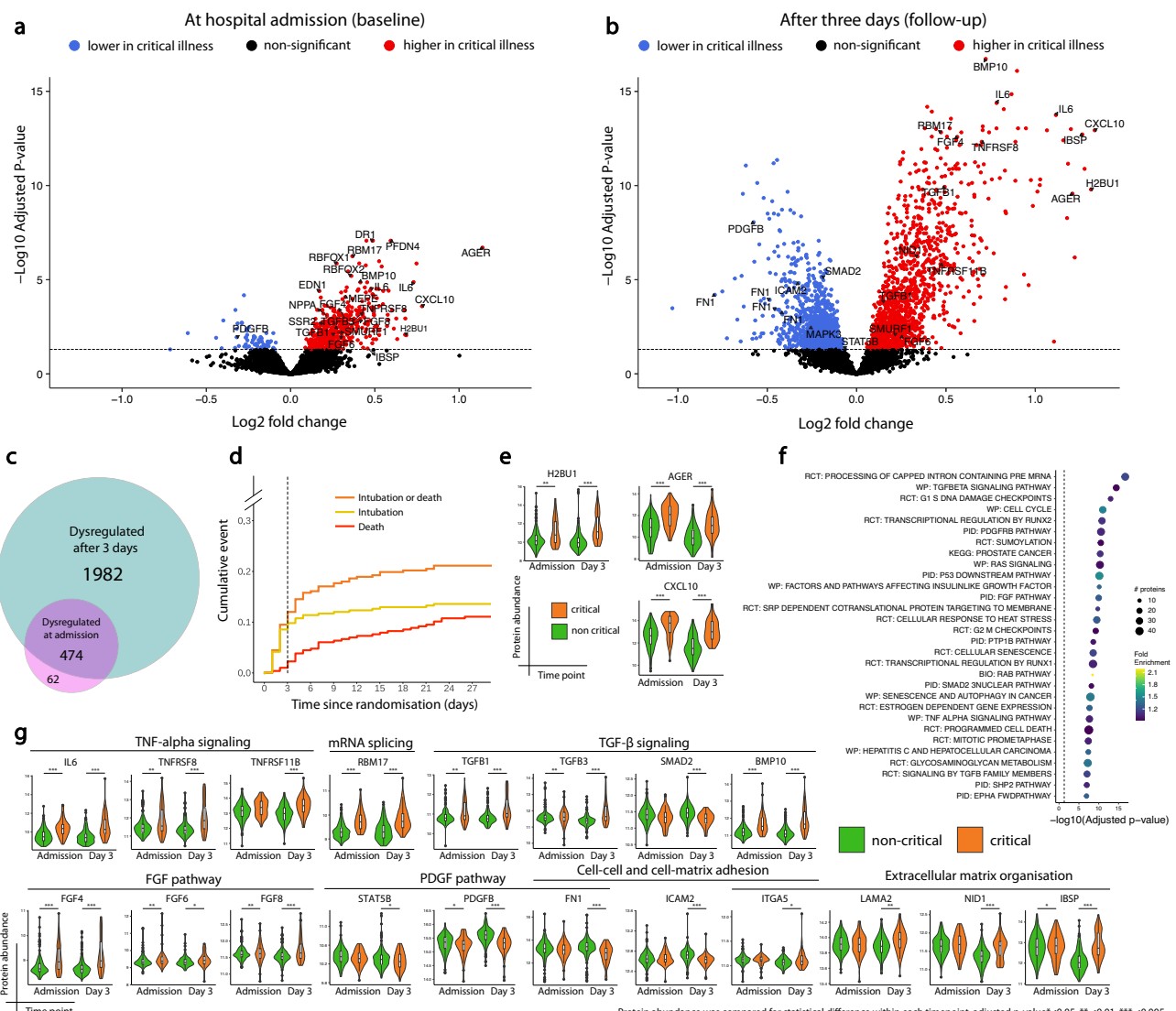

**Fig. 2 | Effect of critical illness on the plasma proteome.** Volcano plots of differentially abundant proteins in patients with non-critical and critical illness at hospital admission (**a**) and 3 days thereafter (**b**). Log2 fold change and significance levels were calculated using linear mixed models. *P* values were adjusted using the Benjamini-Hochberg method. **c** Euler diagram showing the number of proteins with a different protein abundance at hospital admission and 3 days thereafter.
**d** Kaplan–Meier curves of time to intubation, death and composite endpoint of intubation or death. **e** Violin plots of log2 transformed abundance of proteins with the largest fold change in protein abundance by critical illness. Estimates and their confidence intervals were derived from linear mixed models. Upper limit of the violin maximum, lower limit = minimum. For boxplots, centre = median, upper bound = upper quartile, lower bound = lower quartile. Protein abundance was compared for statistical difference within each timepoint, Benjamini Hochberg adjusted *p* values *<0.05, ** <0.01, *** <0.005. *n* = 480 samples from 318 individuals.

**f** Dotplot of enriched pathways 3 days after hospital admission identified by PathfindR (follow-up). The 30 most significant clusters are represented by the most significant pathway term of each cluster. Dot size indicates the number of proteins, colours represent fold enrichment. Terms are ordered by significance level. **g** Violin plots of log2 transformed abundance of example proteins identified from pathways of the most significant clusters shown in (**f**) or pertinent biological processes of respiratory failure. Estimates were derived from linear mixed models. Upper limit of the violin = maximum, lower limit = minimum. For boxplots, centre = median, upper bound = upper quartile, lower bound = lower quartile. Protein abundance was compared for statistical difference within each timepoint, Benjamini Hochberg adjusted *p* values *<0.05, ** <0.01, *** <0.005. *n* = 480 samples from 318 individuals. Source data are provided in the supplementary Source Data file. Full protein names, estimates and *p* values are provided in the Supplementary data 2 (a,e,g), 3 (b,e,g) and 10 (f).

cohort and in the MGH cohort. While only five out of nine proteins were available, the accuracy was even higher (AUC = 0.838) than in our internal validation cohort. Notably, the performance of the five-protein signature within our internal validation cohort exhibited an AUC of 0.802. These findings underscore the parallel performance of our protein signature set in external study cohorts. Furthermore, a five-protein signature set exhibited only marginally reduced performance compared to the nine-protein signature set within our internal validation cohort.

## Effects of imatinib on plasma protein abundance

Last, we assessed the effect of imatinib treatment on the plasma proteome of patients with COVID-19. Despite some imbalances in clinical variables at baseline between placebo- and imatinib-treated patients (Table S5), there was no significant difference in the abundance of any protein at hospital admission, before the start of imatinib or placebo administration (Fig. S11, Supplementary data 6). Imatinib lowered the abundance of 39 proteins and increased the abundance of 53 proteins relative to the placebo group (Figs. 5a, S12, Supplementary data 7).

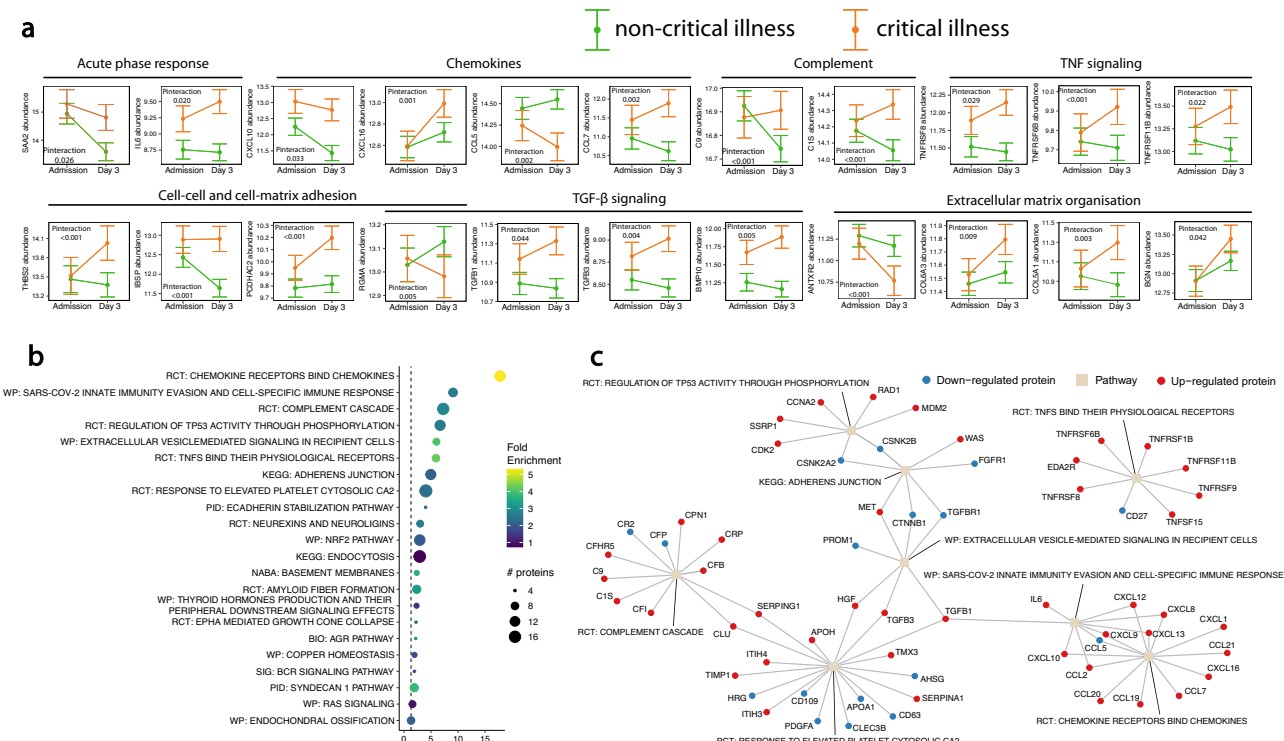

**Fig. 3 | Proteins display early severity-dependent trajectories. a** Line plots of proteins with a different temporal trajectory in patients who developed critical versus non-critical illness. Example proteins were identified using significantly enriched pathways or pertinent biological processes of respiratory failure. Estimated marginal means and their confidence intervals were derived from the linear mixed models using the emmeans::emmeans function. The dots represent estimates, and the error bars indicate the confidence intervals at each time point. Protein abundance levels were Log2 transformed. $n = 480$ samples from 318 individuals. **b** Dotplot of enriched pathways involved in proteins with a different temporal trajectory in patients with a non-critical and critical illness. The 30 most significant clusters are represented by the most significant pathway term of each

cluster. Dot size indicates the number of proteins, colours represent fold enrichment. Terms are ordered by the significance level of the representative pathway term. BIO = Biocarta, KEGG = Kyoto Encyclopedia of Genes and Genomes, PID = Pathway Interaction Database, RCT = Reactome, SIG = Signalling Gateway, WP = Wikipathways. **c** Network plot of the eight most significantly enriched representative pathway terms of the enrichment analysis shown in (**b**). Proteins are connected with each of the pathway term they are involved in. KEGG = Kyoto Encyclopedia of Genes and Genomes, RCT = Reactome, WP = Wikipathways. Source data are provided in the supplementary Source Data file. Full protein names, estimates and *p* values are provided in the Supplementary data 4 (a) and 11 (b).

The two proteins with the highest negative fold change were leucocyte cell-derived chemotaxin-2 (LECT2) and thrombospondin-2 (THBS2; Fig. 5b). The two proteins with the highest positive fold changes were ADP-ribosylation factor-like GTPase 9 (ARL9) and apolipoprotein D (APOD; Fig. 5b). Enrichment analysis revealed that the proteins affected by imatinib were mainly related to ECM organisation, focal adhesion, glycosylation and Ras and TNF signalling (Fig. 5c, Supplementary data 13). A number of proteins was both dysregulated over time in patients who developed critical illness and were modulated by imatinib treatment (Table 2). These proteins include collagens, repulsive guidance molecule A (RGMa), thrombospondin-2, IL-6, biglycan and urokinase-type plasminogen activator (PLAU). These proteins are mediators of virus-induced inflammation, tissue remodelling, extracellular matrix organisation, cell–cell and cell-matrix adhesion[19–26]. These findings indicate that treatment with imatinib in the early phase of hospital admission attenuates perturbations in pathways associated with critical illness, including the acute phase inflammatory response as well as cell adhesion and extracellular matrix turnover.

**Lung and cell-specific attribution of proteins altered by imatinib treatment**

To assess whether modulation of plasma proteins by imatinib reflect changes in the pulmonary compartment, we performed lung and cell-specific attribution analyses using 1. lung transcriptomic data of SARS-CoV-2 infected Syrian hamsters treated with imatinib[27] and 2. an

external COVID-19 tissue atlas of single-cell and single-nucleus RNA sequencing (sc/snRNA-Seq) on the human lung in severe COVID-19[28]. First, of a total of 16.391 genes, imatinib treatment resulted in the upregulation of 866 genes and downregulation of 900 genes in the lung transcriptome of SARS-CoV-2 infected Syrian hamsters. Genes involved in extracellular matrix turnover and cell adhesion that were attenuated by imatinib include thrombospondin-1 (*THBS1*), integrins (e.g. *ITGA5, ITGA8, ITGAX*), collagens (e.g. *COL5A3, COL8A2 and COL27A1*), matrix metalloproteinases (e.g. *MMP9, MMP16 and MMP25*) and proteoglycans (e.g. *DCN, LUM and FMOD*; Fig. S13 and Supplementary data 8). Enriched pathways associated with extracellular matrix turnover and cell adhesion comprise focal adhesion, ECM degradation, collagen formation, and the assembly of integrins and proteoglycans (Fig. S14 and Supplementary data 14). The genes and corresponding proteins *PLAU, IL6, RCAN3, FJX1, PPIC and TGFB3* were affected by imatinib in the lungs of hamsters and in human plasma, respectively (Table S6).

Second, we performed cell-specific attribution of the plasma proteins that were affected by imatinib in critical COVID-19 using the cell-specific atlas derived from the lungs of patients who died from COVID-19 (Fig. S15, Supplementary data 15). In pulmonary fibroblasts, the following genes related to extracellular matrix turnover were upregulated: collagens (*COL1A1, COL3A1, COL5A1 and COL6A1*), the proteoglycan biglycan (*BGN*), thrombospondin-2 (*THBS2*) and metalloproteinase inhibitor 3 (*TIMP3*). Additionally, *TIMP3* was strongly

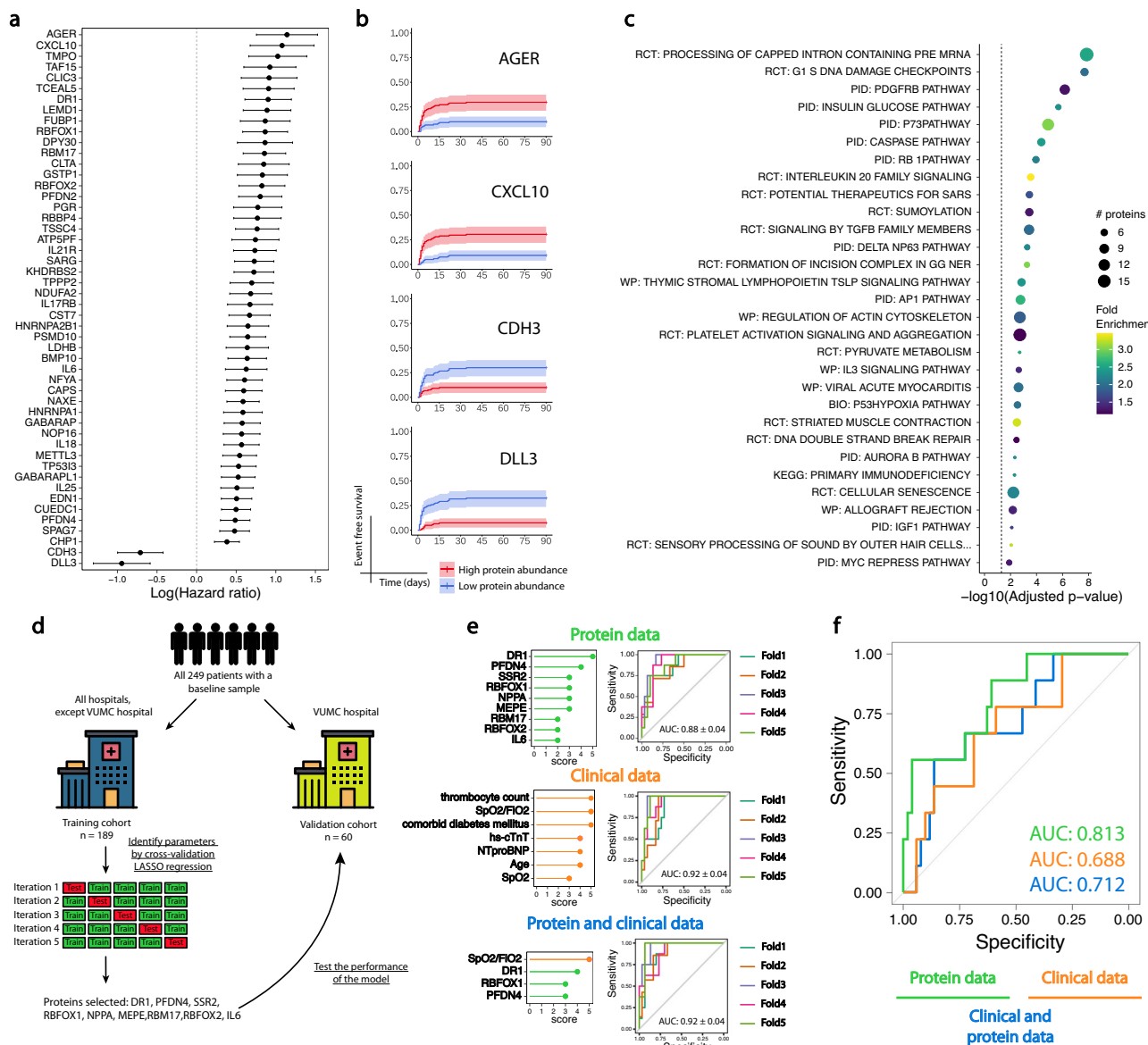

**Fig. 4 | Using the baseline proteome to predict critical illness development.**
**a** Log(hazard ratio) ± the standard error for time to development of critical disease using protein abundance at hospital admission. The 50 proteins with the highest significance levels are shown and ordered from high to low log(hazard ratio). $n = 249$ samples/individuals. **b** Kaplan Meier curves showing the association between protein abundance at hospital admission and time to intubation or death. Protein abundance below the median was considered low, above the median was considered high. The bands denote confidence intervals. The proteins with the two highest and two lowest log(hazard ratio) are shown. $n = 249$ samples/individuals. **c** Dotplot of pathways involved in proteins whose abundance at hospital admission were associated with time to intubation or death, identified by PathfindR. The 30 most significant clusters are represented by the most significant pathway term of each cluster. Dot size indicates the number of proteins, colours represent fold enrichment. Terms are ordered by significance. BIO = Biocarta, KEGG = Kyoto Encyclopedia of Genes and Genomes, PID = Pathway Interaction Database, RCT =

Reactome, WP = Wikipathways. **d** Clinical and protein abundance data at hospital admission were used to predict critical illness development using Least Absolute Shrinkage and Selection Operator (LASSO) regression. The training cohort consisted of all participants except those that were patients at the Amsterdam UMC, location VUmc hospital (VUMC). The model was trained by performing five-fold cross-validation and tested in the validation cohort. **e** Feature importance of proteins and clinical variables based on the LASSO models using protein data (green), clinical data (orange) or both (blue). The performance of the models in the training cohorts is shown in receiver operating characteristic (ROC) curves. The mean area under the curve (AUC) and standard error are given. **f** Performance of the models on the validation cohort using protein data (green), clinical data (orange) or both (blue). The area under the curve (AUC) is given. Source data are provided in the supplementary Source Data file. Full protein names, estimates and $p$ values are provided in the Supplementary data 5 (a,b) and 12 (c).

downregulated in most cells, but upregulated in alveolar type 1 (*AT1*) cells, as well as in lymphatic and vascular endothelial cells. Hevin (*SPARCL1*), involved in extracellular matrix turnover and cell-matrix adhesion[29], was upregulated in cells comprising the pulmonary vascular wall: vascular endothelial cells, pericytes and smooth muscle cells. There was very little differential expression of interleukin 6 (*IL6*) and urokinase-type plasminogen activator (*uPA/PLAU*). Taken

together, these attribution analyses demonstrate how proteins associated with acute respiratory failure are affected in critical COVID-19, and reveal how proteins affected by imatinib in our proteomics analysis are regulated at transcriptome level. Notably, transcriptomics and proteomics are complementary techniques that cannot be directly compared due to the influence of processes such as secretion and degradation, which could introduce disparities in their results.

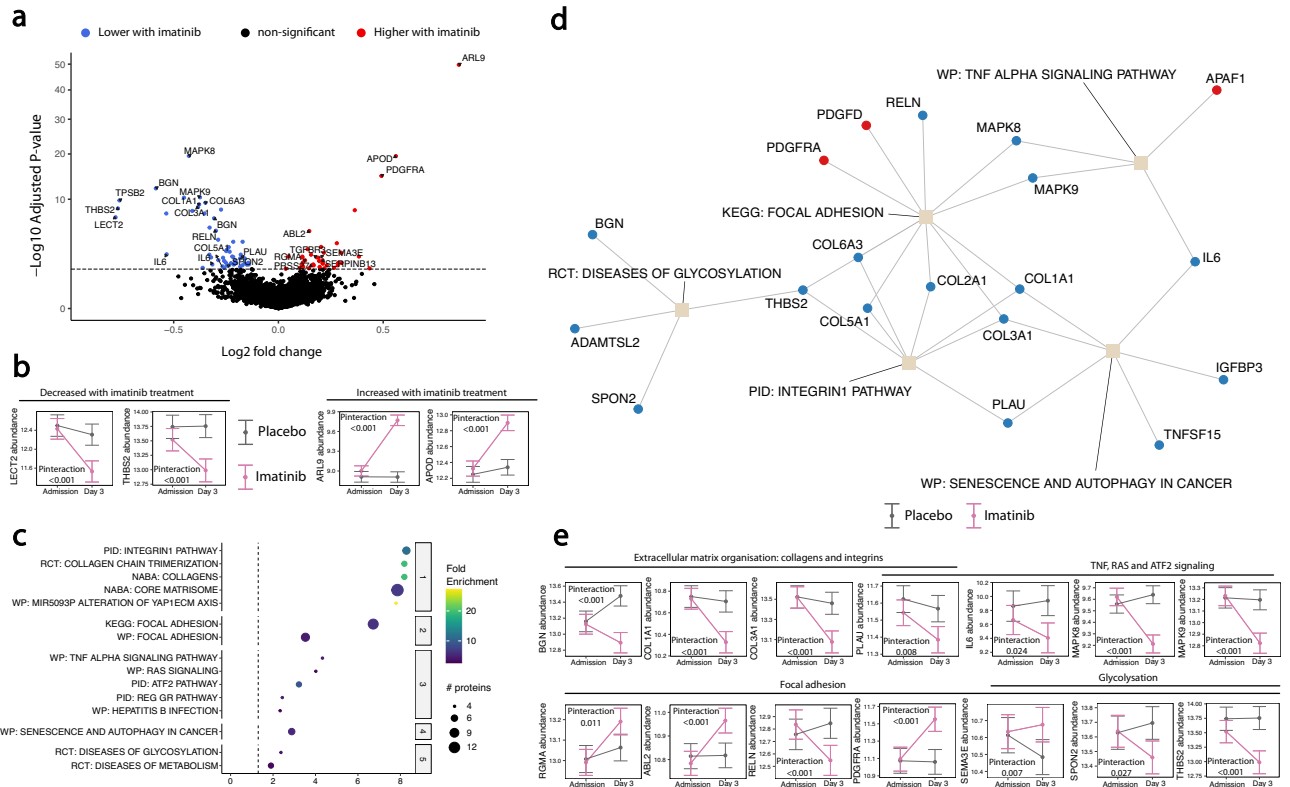

**Fig. 5 | Effects of imatinib on plasma protein abundance. a** Volcano plot of differentially abundant proteins in patients treated with placebo and imatinib. Linear mixed models were applied to calculate the Log2 fold change and significance levels. *P* values were adjusted using the Benjamini-Hochberg method. **b** Line plots of the proteins with the two largest negative and positive fold changes over time between patients treated with placebo and imatinib. Fold changes, estimated marginal means and confidence intervals were derived from the interaction of time and imatinib treatment using linear mixed models. The dots represent estimates, and the error bars indicate the confidence intervals at each time point. *n* = 480 samples from 318 individuals. **c** Dotplot of enriched pathways involved in proteins with a significant interaction of time and imatinib treatment, identified by PathfindR. Up to 5 of the most significant pathway terms of a cluster are shown. Dot size indicates the number of proteins, colours represent fold enrichment. Terms are ordered by significance. BIO = Biocarta, KEGG = Kyoto Encyclopedia of Genes

and Genomes, PID = Pathway Interaction Database, RCT = Reactome, WP = Wikipathways. **d** Network plot of the five most significantly enriched representative pathway terms of the enrichment analysis shown in (**c**). Proteins are connected with each of the pathway term they are involved in. KEGG = Kyoto Encyclopedia of Genes and Genomes, PID = Pathway Interaction Database, RCT = Reactome, WP = Wikipathways. **e** Line plots of proteins with a significant interaction between time and imatinib treatment. Example proteins were identified using significantly enriched pathways or pertinent biological processes. Estimated marginal means and their confidence intervals were derived from linear mixed models. The dot at each time point shows the estimate, the error bars at each time point show the confidence interval. Protein abundance levels were log2 transformed. *n* = 480 samples from 318 individuals. Source data are provided in the supplementary Source Data file. Full protein names, estimates and *p* values are provided in the Supplementary data 6 (b,e), 7 (a), and 13 (c).

## Discussion

In this secondary analysis of a double-blind placebo-controlled multi-center clinical trial, we investigated longitudinal proteomic perturbations associated with critical illness and assessed the effects of imatinib treatment in hospitalised patients with COVID-19. Plasma proteomic analyses were performed using a wide-angle approach that measured 6385 unique proteins at hospital admission and three days thereafter. We found that critical illness was associated with a dysregulation in plasma proteomics of both canonical acute phase response pathways, as well as previously unreported pathways. These identified pathways included, but were not limited to, cell adhesion, extracellular matrix turnover, TGF-β and FGF signalling. Moreover, we found that the plasma proteome at hospital admission provides reliable and reproducible information on critical illness development and prognosis in hospitalised COVID-19 patients. Biomarkers corresponding to dysregulations in extracellular matrix turnover and focal adhesion were attenuated by imatinib treatment in the human plasma proteome. A concurrent mitigatory effect of imatinib on the transcriptome associated with extracellular matrix turnover and focal adhesion was observed in the lungs of SARS-CoV-2 infected hamsters. Taken

together, our study delineates the plasma proteome alterations associated with critical illness in COVID-19 and indicates that attenuation of alveolar-capillary barrier disruption with imatinib treatment is reflected in the plasma proteome. According to other studies that investigated the blood proteome in COVID-19, the most frequently observed and relevant disruptions observed in the proteome of critical illness include: neutrophil degranulation, cell cycle and transcriptional regulation and complement, coagulation, interferon, chemokine and interleukin signalling pathways (i.e. predominantly pathways involved in the acute phase response)[12,13,17,18,30–44]. However, few proteomic studies have identified proteomic profiles associated with tissue injury specifically related to acute respiratory failure[17,38]. In our study, patients from 13 hospitals throughout the Netherlands were systematically followed according to a strict prospective sampling and clinical data registration scheme. The patients in our study comprised a relatively homogenous population, with 92% of patients receiving oxygen supplementation through a nasal cannula or mask at hospital admission. Another unique feature of our study is that we tested the effect of a pharmacological intervention in a randomised setting on pathways associated with critical outcomes. This permitted an evaluation of

**Table 2 | Proteins dysregulated in critical disease and modulated by imatinib**

| Protein | GeneID | Change in critical disease | Imatinib effect |
|---|---|---|---|
| Angiopoietin-related protein 3 | ANGPTL3 | Increase | Decrease |
| Collagen alpha-1(XXVIII) chain | COL28A1 | Increase | Decrease |
| Collagen alpha-1(I) chain:C-term propeptide | COL1A1 | Increase | Decrease |
| Collagen alpha-3(VI) chain:Bovine pancreatic trypsin inhibitor/Kunitz inhibitor domain, isoform 1 | COL6A3 | Increase | Decrease |
| Serine protease 27 | PRSS27 | Decrease | Increase |
| Serpin B13 | SERPINB13 | Decrease | Increase |
| Collagen Type III | COL3A1 | Increase | Decrease |
| Calcipressin-3 | RCAN3 | Increase | Decrease |
| Collagen alpha-1(V) chain | COL5A1 | Increase | Decrease |
| Interleukin-6 | IL6 | Increase | Decrease |
| Luteinizing hormone | CGA\|LHB | Decrease | Increase |
| Tumour necrosis factor ligand superfamily member 15 | TNFSF15 | Increase | Decrease |
| Biglycan | BGN | Increase | Decrease |
| Thrombospondin-2 | THBS2 | Increase | Decrease |
| Urokinase-type plasminogen activator | PLAU | Increase | Decrease |
| Repulsive guidance molecule A | RGMA | Decrease | Increase |
| Apolipoprotein D | APOD | Decrease | Increase |
| Olfactomedin-like protein 3 | OLFML3 | Increase | Decrease |
| Platelet-derived growth factor D | PDGFD | Decrease | Increase |

Proteins that changed over time in patients who developed critical illness and were modulated by imatinib treatment are shown. The direction and significance levels were established in linear mixed models. This was achieved by applying an interaction term between time and critical disease, and time and imatinib treatment, respectively. Benjamini-Hochberg correction was applied to correct for multiple testing with a false discovery rate (FDR) of 0.05. A complete overview of changes over time of alle proteins are presented in Supplementary data SDF4 and SDF7.

whether identified pathways are causally involved in imatinib treatment and critical disease development. Therefore, this study may serve as an example for future studies evaluating the change in biology underlying interventions.

Next to inflammatory dysregulations, the most significant and relevant pathways associated with critical illness in our study can be divided into three main groups according to their functions 1. cell-cell junctions and cell-matrix interaction (as evidenced by proteins regulating focal adhesion and adherens junctions such as integrins, ICAMs, IBSP, RGMa); 2. ECM turnover (e.g. collagens, biglycan, thrombospondin-2, nidogen-1, LAMA4), which are crucial for maintaining structure of alveoli and integrity of the alveolar-capillary barrier function; and 3. FGF (e.g. FGF4 and FGF8) and TGF-β (e.g. TGF-β1, TGF-β3 and BMP10) signalling pathways that regulate tissue remodelling and repair. The alveolar-capillary barrier consists of epithelial cells, a thin collagen-rich basement membrane and a monolayer of endothelial cells[45]. Permeability across epithelial cells is tightly regulated by intercellular tight junctions and adherens junctions that anchor the cytoskeletons of neighbouring cells[46]. In severe COVID-19, the lung's epithelial and endothelial barrier function is compromised resulting in accumulation of protein-rich fluid into the alveoli, a phenomenon that similarly occurs in non-COVID ARDS[25,47,48]. While intercellular gap formation and hyperpermeability are deemed central in the development of inflammatory pulmonary oedema[49], ultrastructural analyses in COVID-ARDS have shown additional and more severe forms of injury to the alveolar-capillary barrier including

detachment and loss of alveolar endo- and epithelial cells as well as turnover of the extracellular matrix[15,50,51]. Alveolar-capillary barrier disruption results in the release of collagens, integrins, nidogens, proteoglycans and laminins from the alveoli into the bloodstream[25,38,52]. Concurrently, macrophages, fibrocytes, fibroblasts and myofibroblasts accumulate in the alveolar compartment, leading to excessive deposition of fibronectin, collagens I and III and other components of the extracellular matrix[53]. This is referred to as the fibroproliferative response, which results in the formation of a provisional ECM[53]. In our study, we found that biomarkers of alveolar-capillary barrier disruption were measurable in plasma and were associated with the development of critical illness. Following alveolar-capillary barrier destruction, repair and regeneration processes such as TGF-β and FGF signalling are required[54–57]. However, these repair processes are often impaired or overactivated in severe viral infections, contributing to the activation of coagulation and fibrosis[54,58,59]. Histopathological analyses confirm that upon alveolar-capillary barrier disruption, the FGF and TGF-β signalling repair mechanisms are impaired in severe COVID-19[15]. In line with previous studies, we observed that in critical illness the plasma proteome reveals enhanced extracellular matrix turnover, impaired intercellular adhesion and inappropriate impair processes[15,38,50–52].

Testing a pharmacological intervention in a randomised setting permitted an evaluation of whether identified pathways are causally involved in imatinib treatment and critical disease development. In the context of profiles associated with critical illness, the most significant processes attenuated by imatinib were 1. TNF signalling and acute phase response (e.g. IL-6, PLAU and CXCL12); and 2. focal adhesion and ECM turnover (e.g. collagens, PLAU, thrombospondin-2 and biglycan). In another secondary analysis of using multiplex assays, we observed that imatinib decreased plasma levels of IL-6, E-selectin and Surfactant protein D (SP-D), indicating attenuation of inflammation, endothelial cell adhesion and epithelial injury[6]. Dampening of inflammation with imatinib treatment in the current study was evident from inhibition of MAPK8, MAPK9, IL-6, CXCL12 and the serine protease urokinase-type plasminogen activator (PLAU or its alias uPA). After binding of uPA to its receptor, uPAR, it converts plasminogen to plasmin which induces a proteolytic cascade resulting in extracellular matrix turnover[60]. Its action is modulated by other serine protease, of which serine protease 27 and serpin B13 levels were suppressed by imatinib treatment in our study[60]. Moreover, IL-6 and uPA are both modulators of vascular permeability and could therefore aggravate vascular leak[61,62]. Proteins with important involvement in extracellular matrix organisation and adhesion include thrombospondin-2 and collagens type I and III, which are most abundant in the alveolar wall and alveolar septa[63]. The effect of imatinib on type 4 collagen, the most abundant collagen in the basement membrane[63], was not measured because no aptamer was available. Thrombospondin-1 and thrombospondin-2 are matricellular proteins involved in multiple processes including cell-cell and cell-matrix interactions, extracellular matrix turnover, wound repair, tissue remodelling and regulation of angiogenesis[24,26]. Thrombospondins interact with integrins, ECM components (e.g. fibronectin, laminin), matrix metalloproteins and FGF and TGF-β signalling[24,26]. In light of our findings of dysregulations preceding critical illness in COVID-19, thrombospondins could play a central role in the alveolar capillary barrier and repair mechanisms. The FGF and TGF-β pathways were not significantly affected by imatinib treatment, although some proteins affected by imatinib have the potential to regulate FGF and TGF-β signalling (i.e. biglycan, thrombospondin-2, RGMA, TGFBR3)[24,26,64,65].

Imatinib was administered in the CounterCOVID trial following earlier observations that it attenuates pulmonary vascular leak under inflammatory conditions in murine models[66–70]. After study completion, it became evident that imatinib also mitigates pulmonary vascular leak and inflammation in rodent models of COVID-19[27,71,72]. Other break point cluster (Bcr) Abelson (Abl) tyrosine-kinase inhibitors (TKI) could

potentially exhibit similar effects on the host response and disruption of the alveolar-capillary barrier. While no clinical studies were conducted with other Bcr-Abl TKIs in patients with COVID-19, beneficial effects of nilotinib[73], ponatinib[74], bosutinib[75] have also been observed in animal models of experimental acute lung injury. In contrast, dasatinib could potentially aggravate pulmonary inflammation and vascular leakage, depending on the dose and model of acute lung injury[76–78]. The divergence in effects on the pulmonary vasculature is most likely explained by the unique profile and potency of each Bcr-Abl TKI for tyrosine kinase targets[79,80]. In line with distinct gene expression of imatinib and bosutinib in endothelial cells associated with vascular quiescence[81], imatinib and bosutinib exhibit low interference, whereas dasatinib, ponatinib, and nilotinib exhibit disturbance of vascular homoeostasis[79,81–84]. Therefore, pulmonary or cardiovascular-related adverse events related to imatinib or bosutinib are generally uncommon[79,85,86]. However, case reports and one case series of 27 patients report that imatinib can induce interstitial lung disease[87–90]. Within the case series, the median time from imatinib initiation to development of interstitial lung disease was 49 days (range: 10–282 days), which was irreversible for 4 patients despite high-dose corticosteroid treatment[87]. Considering the short treatment duration of 10 days in the CounterCOVID trial, development of imatinib-related interstitial lung disease in COVID-19 is unlikely. In the CounterCOVID trial, one patient who received imatinib was diagnosed with 'COVID-19-associated organising pneumonia' based on computed tomography (CT) scan findings[4]. However, we cannot rule out the possibility that these interstitial lung abnormalities were related to imatinib treatment.

Our study has several important limitations. Despite the large number of proteins measured, only those for which an aptamer was available were included. Considering that the human proteome consists of approximately 20.000 different proteins, we could have potentially overlooked important proteins[91]. While proteomics, the Somascan platform in particular, can capture a broad range of proteins and exhibit high reproducibility, proteomic profiling using antibody-based reagents offers a greater number of phenotypic associations and protein target specificity[92]. Notably, a majority (64%) of protein quantitative trait loci (pQTLs) are shared across these platforms[93]. However, inter-platform reproducibility varies strongly for each protein target, with median correlations coefficients between measurements using the Somascan and Olink platforms of 0.38-0.50[92,93]. We compared our Somascan measurements with previously reported data using Luminex (R&D Systems Inc., Minneapolis, United States) multiplex assays on plasma samples from the same patients and time points[6]. Pearson correlation coefficients indicate that some proteins exhibited moderate to strong correlations (IL-6 $\rho$ = 0.80, IL-8 $\rho$ = 0.51 and AGER $\rho$ = 0.74), while others showed very weak or even negative correlations (IL-10 $\rho$ = 0.14, IL-2 $\rho$ = -0.05). These findings suggest that, when applying high throughput proteomics, robust conclusions should be derived primarily from pathway analysis, rather than from the observations of individual proteins. The specificity of four proteins (RBM17, NPPA, MEPE, and IL6) in the predictive nine-protein signature for critical illness development, was confirmed through either singular or multiple assessments, including protein quantitative trait loci (pQTLs) mapping, enzyme-linked immunosorbent assay (ELISA), proximity extension assay (PEA), or bead-based immunoassays[94–100]. Additional confirmation of the specificity of the remaining proteins in the nine-protein signature would therefore benefit its application in future clinical practice. Another limitation is that we only collected plasma samples, as the collection of bronchoalveolar lavage fluid (BALF) or tissue from human patients was either not appropriate or not feasible. The proteome in tissue and BALF is distinct from that in blood and provides more information on biological processes in the bronchoalveolar compartment[101–103]. To gain some insight into the effects of imatinib on the lungs, we contextualised our findings using

RNA sequencing data from extracted lungs of Syrian hamsters who were treated with imatinib after SARS-CoV-2 infection[27]. Pathological manifestations of COVID-19 show many similarities with severe COVID-19 in humans[104], however hamster models can still only be used as a surrogate for humans. Moreover, translation of these effects is restricted by the different study designs. The changes and directions observed in tissue transcriptomics cannot be directly compared to those in plasma proteomics.

Our findings could also have implications in other fields. Patients with critical COVID-19 share many similarities with non-COVID-19 ARDS patients, although COVID-19 ARDS is associated with more alveolar oedema[47]. Impaired TGF-β signalling, ECM turnover and cell adhesion play pivotal roles in alveolar capillary barrier disruption and thus alveolar oedema formation[105,106]. Stratification between non-critical and critical COVID-19 closely approximates stratification between patients who do or do not develop a (COVID-19) ARDS phenotype. The effects of imatinib could also implicate a beneficial effect for the treatment of diseases with increased vascular remodelling such as pulmonary arterial hypertension (PAH). Imatinib modulated a substantial number of proteins that are taking part in the pathobiology of PAH. These include thrombospondin-2, collagens, biglycan and members of the TGF-β superfamily, and are involved enhanced tissue remodelling and extracellular matrix turnover of pulmonary arteries in PAH[107,108].

This study demonstrates that the plasma proteome of patients with respiratory failure due to COVID-19 is characterised by perturbations in inflammation, enhanced extracellular matrix turnover, impaired cellular adhesion and inappropriate remodelling processes. These pathways may well reflect sustained destruction of all components of the alveolocapillary barrier including epithelial and endothelial cells, as well as the basement membrane. We identified a small protein signature that demonstrated accurate risk stratification in both internal and external validation for development of critical illness. Imatinib treatment particularly attenuated dysregulations of extracellular matrix turnover and impaired intercellular adhesion.

## Methods

The design, study protocol, and primary results of the CounterCOVID trial were reported previously[4,5]. Briefly, the CounterCOVID trial was a double-blind placebo-controlled randomised clinical trial conducted in 13 academic or teaching hospitals in the Netherlands. Eligible participants were above 18 years of age, had a reverse transcription polymerase chain reaction (RT-PCR)- confirmed SARS-CoV-2 infection, and were hypoxemic or required supplemental oxygen to maintain a peripheral oxygen saturation above 94%. The exclusion criteria comprised, but were not limited to, pre-existing severe pulmonary disease, pre-existing heart failure and concomitant treatment with medications known to strongly interact with imatinib. After obtaining written informed consent, participants were 1:1 randomised to receive a loading dose of 800 mg imatinib, followed by 400 mg of imatinib once daily or an equivalent number of placebo tablets for a total of 10 days. Randomisation was done with the Castor Electronic Data Capturing System (Castor EDC; Amsterdam, Netherlands). Randomisation was stratified by hospital site using variable block sizes of two, four, or six patients. All patients, health care providers and study investigators were blinded to treatment allocation. The primary study endpoint was time to discontinuation of ventilation and supplemental oxygen for more than 48 consecutive hours, while being alive during a 28-day period after randomisation. Prespecified secondary endpoints involved mortality, duration of both invasive and non-invasive ventilation, length of stay in the intensive care unit, requirement for extracorporeal membrane oxygenation (ECMO) and the longitudinal need for oxygen supplementation[4,5] drug safety[4], plasma host response biomarkers[6] and pharmacokinetics[109,110]. For the purpose of this study, critical illness was defined as the need for invasive

ventilation and/or fatal outcome during the 90-day follow-up period. This trial was conducted in compliance with all relevant ethical regulations, including the guidelines of the International Conference on Harmonization Good Clinical Practice and the Declaration of Helsinki. The trial protocol was approved by the medical ethics committee of the Amsterdam UMC (VUmc, Amsterdam, Netherlands) and institutional review board of the Amsterdam UMC, location VUmc. Study participants did not receive compensation for their involvement.

This study comprises a secondary analysis of both clinical data and plasma samples derived from a clinical trial that was registered with the EU Clinical Trials Register (EudraCT 2020–001236–10, https://www.clinicaltrialsregister.eu/ctr-search/trial/2020-001236-10/NL) and Netherlands Trial Register (NL8491, https://www.trialregister.nl/trial/8491).

## Plasma samples and clinical data acquisition

The data of all patients included in the CounterCOVID trial with at least one plasma sample available during the first three days of treatment were used for this study. Samples were collected during the first 24 h of hospital admission and before the start of treatment with placebo or imatinib (baseline) and 3 days after first study drug administration (Day 3). Peripheral blood samples were collected in lithium heparinized tubes. Within 2 h of collection, samples were centrifuged in the laboratory of the participating hospital and the supernatant was stored in aliquots of 500 µL at −80 °C. If a plasma sample from the third day was not available, the sample from the second day was used instead. Plasma samples were shipped by air transport from the coordinating hospital in Amsterdam to the United States of America (Boulder, Colorado) for proteomic analysis in designated cardboard boxes covered with dry ice. Clinical data were recorded using Castor electronic data capture (Castor EDC) and were monitored by an independent contract research organisation of the Amsterdam UMC.

## Proteomic assays

Proteomics was performed in ~55 µL plasma using aptamer assays by SomaLogic Inc. Proteins were quantified using modified aptamers, which are reagents made of chemically modified DNA sequences that bind to each protein of interest[111]. From 7596 protein targets, 308 non-human protein targets were excluded, leaving 7288 (corresponding to 6385 unique proteins or protein complexes) available for inclusion in the proteome analysis. For some proteins, more than one aptamer is available, to recognise distinct epitopes, isoforms, protein conformations or protein fragments, or a combination of these (detailed in Supplementary data 1). Relative protein abundance in each sample was measured in Relative Fluorescence Units (RFU). Hybridisation control normalisation, intraplate median signal normalisation, and median signal normalisation were performed to remove systematic bias in the assay data. Plate scaling was applied to remove systemic variability between assay plates. Normalisation ratios that exceeded the interval of 0.40–2.50 were flagged. Protein abundance values were Log2 transformed to obtain a more normal distribution.

## Statistical analysis

All statistical analyses were conducted on plasma samples collected from all 318 patients, unless stated otherwise. Baseline characteristics were compared using an unpaired t-test for normally distributed continuous variables, a Mann-Whitney $U$ test for non-normally distributed continuous data and a chi-square test for categorical variables. Quality control was assessed using relative log expression (RLE) plots, assessing the percentage of variance that is explained by each variable of interest, principal component analysis and logistic regression analysis. The RLE plots and assessment of explained variance for each variable of interest were calculated using the plotRLE and getVarianceExplained functions from the Bioconductor scater package[112]. Principal Component Analysis (PCA), linear regression and logistic

regression were performed using the prcomp, lm and glm functions respectively from the R stats package. Body mass index (BMI) measurements were missing for 22 (7%) patients and were imputed by the median. Statistical analyses were performed using R version 4.2.1 and RStudio version 2023.03.0 Cherry Blossom.

## Quality control

The abundance profiles of all samples are visualised using Relative Log Expression (RLE) plots and Principal Component Analysis (PCA) in Figs. S2 and S3. In RLE plots, unwanted variation in the data is visualised which can be particularly useful in detecting normalisation failure[113]. PCA is a dimensionality-reduction method that summarises the variance of data in so called principal components[114]. The RLE plots and PCA indicated that the 55 from 535 samples that were flagged by Somalogic contain aberrant results compared to samples that were within the normalisation ratio interval. Logistic regression analysis showed that neither clinical, patient characteristic, laboratory findings nor medication use could explain flagging of the samples (Table S2). However, samples that were flagged, were approximately 18 times more likely to have a low volume or caused clogging during processing ($p < 0.001$, Table S2). Since the accuracy of the flagged samples could not be guaranteed, and these samples most likely contains deviant information for technical reasons, these 55 samples were removed from further analysis. The getVarianceExplained function from the scater package was used to compute the percentage of variance that is explained by variables that could possibly generate bias. Out of the variables of interest, hospital site and flagging of the samples explained a significant amount of variance, respectively 6.56% and 6.47% (Fig. S4 and Table S2). In univariate linear models, the abundance of all proteins at baseline was similar between patients assigned to placebo and imatinib (Supplementary excel file SE1).

## Differential protein abundance

For differential protein abundance analysis, linear mixed models were performed using the lmer function from the lmerTest package[115]. Restricted maximum likelihood (REML) calculated P-values using a type 3 F-test, in conjunction with Satterthwaite's method for estimating the degrees of freedom for fixed effects were fit[115]. Benjamini-Hochberg correction was applied to correct for multiple testing with a false discovery rate (FDR) of 0.05. We used linear mixed models to investigate the difference between patients with critical versus non-critical illness. The following covariates were included: time, critical illness, time and critical illness interaction, imatinib treatment, hospital site, age, biological sex, BMI, comorbid diabetes mellitus and cardiovascular disease. A random intercept was given to each patient to account for individual effects. The linear mixed model formula in Wilkinson-style notation was specified as follows:

Protein abundance ~ time + critical illness + time × critical illness + treatment + hospital site + age + biological sex + BMI + diabetes mellitus + cardiovascular illness + (1| patient)

The effect of treatment on relative protein abundance was investigated using linear mixed models with time, the interaction of time and treatment, hospital site and a random intercept for each patient. The linear mixed model formula in Wilkinson-style notation was specified as follows:

Protein abundance ~ time + time × treatment + hospital site + (1| patient)

## Critical illness: time-to-event analysis

The association between protein abundance at baseline and time to intubation or death was investigated using Cox proportional hazards regression analyses with hospital site, age, biological sex, BMI, and comorbid diabetes mellitus and cardiovascular disease as covariates. This was performed using the coxph function from the survival package. To make this analysis easier to interpret and more applicable, data

were centred and scaled, so that the mean was 0 and the standard deviation was 1. In this way, deviation from the mean, rather than relative protein abundance was used to estimate the risk for critical illness development. Benjamini-Hochberg correction was applied to correct for multiple testing with a false discovery rate (FDR) of 0.05.

## Pathway enrichment

The R package pathfindR (v1.6.4) was used for pathway enrichment analyses[116]. PathfindR identifies active subnetworks of significant proteins, and subsequently applies hierarchical clustering of enriched terms. In contrast to many other enrichment methods, pathfindR takes into account significance levels of each differentially abundant protein. Protein sets were defined using the Molecular Signatures Database (MSigDB) C2 canonical pathways[117]. The pathfindR package incorporates information from a protein-protein interaction network (PIN) to enhance pathway enrichment results. In order to adjust for the statistical background during hypergeometric-distribution-based tests in enrichment analysis, we modified the BioGRID PIN (version 4.4.211) to include only the 7288 protein targets measured by the SomaScan assays[118]. Enriched pathways were grouped using hierarchical clustering with the pathfindR::cluster_enriched_terms function. Pathways that were significant (Benjamini-Hochberg adjusted $p$ value < 0.05) in at least 25 out of 50 iterations and contained more than three differentially abundant proteins were included in the final results. From each of the 30 most significantly enriched clusters, the most significantly enriched pathway terms, i.e. representative terms, were selected and visualised together in a dotplot.

## Predictive value of protein abundance at baseline

To identify the proteins with the highest predictive value at baseline for developing critical illness, least absolute shrinkage and selection operator (LASSO) regression was applied using the caret and glmnet packages[119,120]. The training cohort consisted of patients from all but one hospital, the Amsterdam UMC location Vrije Universiteit Medical Center (VUMC), which made up approximately 20% of the full cohort and constituted the validation cohort. The data were centered and scaled prior to model training. The model was trained by performing five-fold cross-validation within the training cohort. Specifically, the lambda value that maximised the mean area under the receiver operating characteristic curve (AUC) during cross-validation was selected. A final model (with the selected lambda parameter) was fitted to the entire training cohort, before being tested in the validation cohort. Feature importance was subsequently defined as the number of models in which each feature had a non-zero coefficient.

To assess generalisability, we externally validated our models by leveraging two previously conducted, independent proteomics studies that provide open access to their data. The selection of these studies was based on the concordance in measured proteins and/or study design. Both studies employed the Somascan platform to evaluate protein abundance. The first study enroled 54 patients with end-stage kidney disease (ESKD) who contracted COVID-19 and were admitted to the Hammersmith Hospital (HH) in London, United Kingdom[18]. All nine proteins were available for the prediction models, whereas only 2 out of 7 clinical variables (comorbid diabetes mellitus status and age) were available[18]. The second study was conducted at the Massachusetts General Hospital (MGH) in Boston, United States of America, and assessed the abundance of 4776 unique proteins or protein complexes in 308 patients with COVID-19 upon hospital admission[17]. An earlier version of the Somascan assay was employed, resulting in the inclusion of only five out of nine proteins from our signature set. From the clinical parameters, only comorbid diabetes mellitus status was available[17]. Additionally, the performance of the model on the MGH cohort was compared to the performance in our own validation cohort using the same five proteins.

## RNA-sequencing on extracted lungs of Syrian hamsters

To contextualise our findings related to the effect of imatinib in the human plasma proteome, we re-analysed RNA-sequencing data on extracted lungs of SARS-CoV-2 infected Syrian hamsters[27]. This dataset was retrieved on request from a previously conducted study by Xia et al. in which Syrian hamsters were intranasally infected with a SARS-CoV-2 B.1.351 mutation virus strain and treated with intragastric imatinib (15 mg/kg) or mock during the first four days following infection. The hamsters were euthanised and sacrificed 7 days after infection, and their lungs were extracted for RNA-seq and pathological analysis. The full study protocol is provided elsewhere[27]. The Syrian hamster (*Mesocricetus auratus*) assembly and gene annotation from the Ensembl genome browser were used to convert hamster geneIDs to human geneIDs using the getLDS function from the biomaRt package. Lowly expressed genes were filtered with the filterbyEpxr function and trimmed mean of M-values (TMM) normalisation was performed using the edgeR package[121]. Negative binomial generalised log-linear models were applied for differential expression using the glmLRT function from the edgeR package[121]. Pathway enrichment analyses were performed with pathfindR, with the same methods as were used for the human plasma proteome pathway enrichment.

## Deconvolution of RNA-sequencing analysis in human lung tissue

External data from a COVID-19 tissue atlas[28], specifically derived from single-cell and single-nucleus RNA sequencing (sc/snRNA-Seq) analyses, were employed to evaluate genotypic alterations across various cell types within pulmonary tissue samples obtained from patients with fatal COVID-19. Using this dataset, we conducted a deconvolution analysis to investigate how genes encoding proteins whose abundance was influenced by imatinib treatment and were enriched in our pathway analyses (as observed in our proteomics analysis), were regulated at the RNA level across various cell types within the lungs of patients experiencing critical illness. Genes demonstrating significant differential expression between healthy lungs and lungs of patients who died from COVID-19 underwent refinement guided by proteins that both displayed significant time and treatment interactions and which were enriched in the pathway analysis as observed in our proteomics analysis. The magnitude (log2 fold change) of the differential expression in each cell between lung tissue of healthy and COVID-19 patients are visualised in a clustered heatmap using the pheatmap function (Fig. S15).

## Reporting summary

Further information on research design is available in the Nature Portfolio Reporting Summary linked to this article.

# Data availability

Pseudonymized source data utilised in this study and generated data are publicly available without restriction. The raw data required for deconvolution of RNA-seq analysis in human lung tissue can be obtained from DOI: 10.1038/s41586-021-03570-8.

For this study, external RNA-seq data from SARS-CoV-2 infected hamsters was retrieved from https://doi.org/10.1038/s41423-023-00985-3. SomaScan proteomics data from ref. 17 can be accessed from https://doi.org/10.1016/j.xcrm.2021.100287. SomaScan proteomics data from ref. 18 can be accessed from https://doi.org/10.1038/s41467-022-35454-4.

Qualified researchers can request access to additional pseudonymised data and related study documents by contacting Erik Duijvelaar at e.duijvelaar@amsterdamumc.nl, affiliated with Amsterdam University Medical Centers, Department of Pulmonary Medicine, Amsterdam Cardiovascular Sciences, Amsterdam, Netherlands Requests will be subject to review, and data will only be shared in accordance with applicable privacy regulations and ethical

considerations. Identifying information will be carefully removed to safeguard the confidentiality of study participants. Please allow up to five working days to process and respond to your data access request. Source data are provided with this paper.

## Code availability

The R script required to reproduce the analyses is available from Zenodo (https://doi.org/10.5281/zenodo.10142777). The code is publicly available and can be reused without restriction.

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

## Acknowledgements

This work was supported by an unrestricted grant from the Amsterdam Medical Center Foundation, acquisition by H.B., a Bottom-up grant from Nederlandse organisatie voor Wetenschappelijk Onderzoek (NWO) ZonMW (grant number 10430 01 201 0007), acquisition by J.A. and H.B. and Innovative Medicines Initiative 2 Joint Undertaking (grant number 101005142), acquisition by J.A. The funding sources had no role in the design of the study, data collection and analysis, or decision to submit the paper for publication. We would like to thank Dr. Lin Xia, Prof. Wen Liu, Prof. Ning-shao Xia and their research team for generously sharing their data on the pulmonary transcriptome of SARS-CoV-2 infected hamsters. This data was critical for our research, and we appreciate the effort and resources that went into it. The protein structure displayed in Fig. 1 was retrieved from https://alphafold.ebi.ac.uk/entry/Q2MGZ8 and was vectorized using Adobe Illustrator 2023. All other figures incorporated into the manuscript figures have been either created by the

authors or obtained under license. Collaborators on behalf of the CounterCOVID Study Group: Sara Azhang, Imke H Bartelink, Ahmed A Bayoumy, Pierre M Bet, Siebe G Blok, Wim Boersma, Peter I Bonta, Karin A T Boomars, Lieuwe D J Bos, Liza Botros, Job J M H van Bragt, Gert-Jan Braunstahl, Lucas R Celant, Katrien A B Eger, J J Miranda Geelhoed, Yurika L E van Glabbeek, Hans P Grotjohan, Laura A Hagens, Chris M Happe, Boaz D Hazes, Leo M A Heunks, Michel van den Heuvel, Wouter Hoefsloot, Rianne J A Hoek, Romke Hoekstra, Herman M A Hofstee, Nicole P Juffermans, E Marleen Kemper, Azar Kianzad, Renate Kos, Peter W A Kunst, Ariana Lammers, Ivo van der Lee, E Laurien van der Lee, Anke H. Maitland-van der Zee, Frances S de Man, Adinda Mieras, Mirte Muller, Elisabeth C W Neefjes, Anton Vonk Noordegraaf, Esther J Nossent, Laurien M A Oswald, Maria J Overbeek, Carolina C Pamplona, Nienke Paternotte, Yigal Pinto, Niels Pronk, Michiel A de Raaf, Bas F M van Raaij, Merlijn Reijrink, Job R Schippers, Marcus J Schultz, Ary Serpa Neto, Elise M A Slob, Patrick J Smeele, Frank W J M Smeenk, Marry R Smit, A Josien Smits, Janneke E Stalenhoef, Pieter R Tuinman, Arthur L E M Vanhove, Jeroen N Wessels, Jessie C C van Wezenbeek

## Author contributions

E.D. acquired the data and takes responsibility for the integrity of the data. E.D. and J.G. performed the statistical analyses. E.D., J.A., J.G., J.P. and H.B. were involved in data interpretation and drafting of the manuscript. All authors provided critical feedback on the manuscript and approved the final version. All authors had full access to the study data and took responsibility for the decision to submit the manuscript for publication.

## Competing interests

J.A. is a co-inventor of a patent (WO2012150857A1, 2011) covering protection against endothelial barrier dysfunction through inhibition of the tyrosine kinase Abl-related gene (Arg). All other authors have no competing interests to declare.
