## [Peer Review File · Nature Communications]

REVIEWER COMMENTS

Reviewer #1 (Remarks to the Author):

I have reviewed the manuscript. This is a very nicely performed study and a well written paper. I have a few comments/suggestions.

1) As shown previously, imatinib therapy can be associated with pulmonary toxicity including interstitial lung disease, even in the short-term use. So, I think this needs to be discussed in the discussion.

2) So, what do the authors think? Are these effects of imatinib drug-specific or similar results can be accepted with the use of other BCR::ABL1 TKIs? I think the authors should comment on this (e.g., are there any in vitro/in vivo data on the use of other TKIs in COVID-19 pneumonia?).

Reviewer #2 (Remarks to the Author):

Nature comms Duijvelaar E et al. 2023

This manuscript is a very well written and crafted manuscript. The authors did longitudinal proteomic profiling of plasma from 318 COVID19 patients to investigate the effects of critical COVID19. They also assess the effects of imatinib. There was some in vivo hamster experiments to describe these effects in the lungs.

Within this research are some really interesting findings and with the deep phenotyping, longitudinal sampling and treatment experiments the project should provide a wealth of new information that can form the basis of clinical based decisions in the management of COVID-19 and other respiratory viral infections. However, my main criticism is not at the experimental design, execution or data analysis. This was all of a pretty high standard. I think on occasion however, the authors would be better off not stating as facts certain assumptions that are implied by the data. The two main examples are the use of the hamster model which the RNAseq data indicates similarities between the human plasma proteome and the hamster lung transcriptome. The conclusion the authors draw is this proves thus that the differential expression shown in human plasma is derived from the pulmonary compartment. This would seem to be a reasonable suggestion, but you are looking at correlation at best. It does not prove that all these changes are derived from lung. For a start, the authors didn't look at any other hamster tissue transcriptomes and looking at some of the proteins these are clearly derived from other tissues such as

liver but found in the circulatory system e.g. ApoD (which you demonstrate is upregulated on treatment with Imatinib).

Secondly, and I feel the association here is a bit stronger, but I'm not convinced that your data clearly proves that you have alveolar-capillary barrier disruption which is then reversed by Imatinib. I think you have grounds to state that the observed molecular changes are indicative of alveolar-capillary barrier disruption but I'm not convinced you have demonstrated absolute proof. It may feel pendantic the difference, but I think it is important that we recognise the hierarchy of evidence here.

I would like to see a revised title and certain parts of the text where these fundamental absolutist statements are watered down. I personally don't think it will harm the paper because it is a well executed project.

Some minor issues

Line 101 – aberrant protein expression levels?? Figure S1 doesn't really tell you anything about that?

Line 103 – randomly assigned – how? Were you able to maintain age and sex match?

Line 156 – “clinical deterioration is reflected by temporal changes in the plasma proteome.

Line 187- I think 4a is showing a subset of these proteins. I think that needs explaining in the text.

Line 234. – the Referred to Figure S10 doesn't really show anything

Line 238 – 5A should be 5B

Figure 5 B what is n?

Role of comorbidities should be underplayed as they are significantly different in your cohort

Line 256 – surplus comma

I think the data interrogation of the hamster data does not necessarily prove that this is a lung-isolated mechanism in humans.

Line 476-486 What volume of plasma is used per assay?

Reviewer #3 (Remarks to the Author):

In this original research manuscript, the authors analyzed serum proteomic data from 385 patients from a clinical trial in which imatinib had been tested as a therapy for COVID-19. They found that most proteins increased at enrollment among subjects who developed critical illness proteins remained increased levels at day 3, and that certain proteins followed disease-outcome-specific trajectories. LASSO-based regression identified a group of proteins that predicted the subsequent development of critical illness. Surprisingly, there was only a modest effect of imatinib on plasma proteins. Overall, the study was well-designed and the analyses follow reasonable and appropriate statistical approaches, although an external validation cohort would have been preferable. Recognizing these strengths, the size/scope of the study and the challenge of such a study during the pandemic, it is a bit less clear what specific new insights into SARS-CoV-2 pathobiology or severe critical illness the results add beyond extensive literature on both of these topics – specifically, the principal assertion that = disruption of the alveolar-capillary barrier is related to disease severity/progression is well-aligned with the current paradigm of acute lung injury/ARDS, albeit of somewhat limited novelty.

Major comments:

Many of the violin/box plots are missing statistical annotation.

The use of bulk lung tissue RNA-seq data from Syrian hamsters infected with SARS-CoV-2 really provides little added mechanistic insight as there seemed to be modest concordance with imatinib-interacting proteins. If a “cellular attribution”-type deconvolution analysis was sought, it could be performed using single-cell data from SARS-CoV-2 tissue atlases (for example PMID 33915568, 33915569).

The study would be considerably strengthened by external cohort validation of the predictive protein signature.

Minor comment:

Throughout the text, the terms upregulated/downregulated and or differentially-expressed are used to describe proteins with different abundance – it is a minor point, but these differences likely do not result from a regulated process or differences in expression, but rather leak into circulation from injured/dying cells – the text should be clarified to more precisely note differential abundance or increased/decreased levels.

The discussion is a bit lengthy and speculative, this could be shortened.

Reviewer #4 (Remarks to the Author):

In this paper the authors report results from a proteome analyses using aptamer technique (6385 plasma proteins measured) in blood from patients in an RCT to test how imatinibe affects outcome in hospitalized patients with Covid 19. The proteome is assessed at hospital admission and 3 days later.

The paper has several strengths including the relatively large sample size, serial sampling and the randomized intervention with a tyrosine kinase inhibitor. Thus, the study does not only inform on proteome perturbations in Covid 19 and critical illness but also how imatinibe affects the proteome.

I find the analyses well performed and the paper well presented and written. There is transparent reporting of QC procedures and results.

A weakness that is inherent to most studies using explorative proteomics is the difficulties in interpretation of the results. A number of positive signals are reported that suggest regulation of certain networks, on the contrary, critical proteins that are not regulated are difficult to interpret and sometimes underreported. There is also a lack of conformity between different methodological platforms used and aptamer based studies are not always supported by antibody techniques.

Questions and comments:

It is unclear to me when plasma samples were retrieved in relation to Imatinib. Were plasma samples retrieved 3 days after starting Imatinib or three days after inclusion? This is crucial for interpretation of the data and should be clarified.

I find it strange that the “clinical + protein model” performs worse than the “protein-alone model” when tested in the validation cohort. Giving the names (and number) of the proteins included in the “clinical+protein model” versus the “protein alone model” could clarify and show that other proteins were selected.

Which clinical variables were used for training (same as in Table 1?) and what variables were selected for prediction models?

Include a statement that “aptamer based proteomics was used” in the last paragraph of the introduction. This is central information for the reader.

Was there a rationale for selecting aptamer based techniques (somascan) in favor of antibody techniques (OLINK)? Please comment in the text of the likelihood of being able to reproduce the results using other platforms? (see Katz et al., Sci. Adv. 2022) This could be especially relevant for the prediction model based on 9 proteins.

The sentence is incomplete: Page 3, p 83-83: “Identification of pathways that contribute to critical illness may highlight novel biomarkers that could be used risk for risk stratification and provide targets for the treatment of respiratory failure in 85 COVID-19 and acute respiratory distress syndrome (ARDS).”

Missing data on BMI was imputed. Where no other proteindata imputed?

Point-by-point response to Reviewers' comments:

Reviewer #1 (Remarks to the Author):

I have reviewed the manuscript. This is a very nicely performed study and a well written paper. I have a few comments/suggestions.

RE: We appreciate the time and effort the reviewer put into reviewing the manuscript.

1) As shown previously, imatinib therapy can be associated with pulmonary toxicity including interstitial lung disease, even in the short-term use. So, I think this needs to be discussed in the discussion.

RE: We have incorporated the imatinib related interstitial lung abnormalities in the discussion section (Line 600). Specifically, we merged this discussion with our response to the next comment, forming a distinct paragraph within the discussion section.

2) So, what do the authors think? Are these effects of imatinib drug-specific or similar results can be accepted with the use of other BCR::ABL1 TKIs? I think the authors should comment on this (e.g., are there any in vitro/in vivo data on the use of other TKIs in COVID-19 pneumonia?).

RE: Here, we think it is important to distinguish the effects of BCR-Abl TKIs on the integrity of the alveolar-capillary barrier, viral replication and on the host response during virus-induced inflammation. In general, these effects are imatinib specific, albeit that some effects are also observed with other BCR-Abl TKIs.

With regard to the effects of various BCR-Abl TKIs on vascular permeability, data are available for the following BCR-TKIs: imatinib, bosutinib, dasatinib, ponatinib, and nilotinib. Both bosutinib and imatinib exhibit high inhibitory activity against Abelson related gene (Arg/Abl2)⁹, which holds a pivotal role in inflammatory vascular leakage^{10,11}. Additionally, favourable effects on inflammatory vascular leak have been observed with nilotinib¹², ponatinib¹³ and bosutinib⁹ in animal models of experimental acute lung injury. In contrast, dasatinib was shown to disrupt vascular integrity, especially in higher doses. Therefore, the protective effect of BCR-Abl TKI seem to be dependent on the protein targets profile. Regarding the effects on the host immune response, imatinib is recognized for its immunomodulatory properties in patients with COVID-19⁶. Conversely, the effects of other Abl TKIs on the host immune response are not as well established. Nevertheless, bosutinib⁹, dasatinib¹⁴ and ponatinib¹³ attenuate IL-6 levels in bronchoalveolar lavage (BAL) fluid in experimental vascular injury studies involving rodents. Considering the central role that IL-6 plays in COVID-19 related inflammation, it is plausible that other BCR-Abl TKIs may also interact with the host immune response. However, this is very speculative, and is not substantiated through preclinical or clinical investigations of COVID-19. With regard to viral replication, imatinib turned out as a top tier compound in a library screen to inhibit SARS-CoV-2 entry in a lung organoid model¹⁵. As this compound library also included other BCR-Abl TKIs (bosutinib and nilotinib), this effect seems specific for imatinib.

The text below has been added in response to both suggestions made by the reviewer on line 551 of the revised manuscript:

“Imatinib was administered in the CounterCOVID trial on the premise that it attenuates pulmonary vascular leak under inflammatory conditions in murine models^{10,16–19}. After study completion, it became evident that imatinib specifically mitigates pulmonary vascular leak and inflammation in rodent models of COVID-19^{20–22}. Other break point cluster (Bcr) Abelson (Abl) tyrosine-kinase inhibitors (TKI) could potentially exhibit similar effects on the host response and disruption of the alveolar-capillary barrier. While no clinical studies were conducted with other Bcr-Abl TKIs in patients with COVID-19, beneficial effects of nilotinib¹², ponatinib¹³, bosutinib⁹ have also been observed in animal models of experimental acute lung injury. In contrast, dasatinib could potentially aggravate pulmonary inflammation and vascular leakage, depending on the dose and model of acute lung injury^{23–25}. The divergence in effects on the pulmonary vasculature is most likely explained by the unique profile and potency of each Bcr-Abl TKI for tyrosine kinase targets^{26,27}. In line with distinct gene expression of imatinib and nilotinib in endothelial cells associated with vascular quiescence²⁸, imatinib and bosutinib exhibit low interference, whereas dasatinib, ponatinib, and nilotinib exhibit disturbance of vascular homeostasis^{26,28–31}. Therefore, pulmonary or cardiovascular-related adverse events related to imatinib or bosutinib are generally uncommon^{26,32,33}. However, case reports and one case series of 27 patients report that imatinib can induce interstitial lung disease^{34–37}. Within the case series, the median time from imatinib initiation to development of interstitial lung disease was 49 days (range: 10–282 days), and this was irreversible for 4 patients despite high-dose corticosteroid treatment³⁴. In the CounterCOVID trial, one patient who received imatinib developed interstitial lung fibrosis during the 28-day follow-up period, versus none of those who received placebo⁴. A diagnosis of 'COVID-19-associated organizing pneumonia' was based on computed tomography (CT) scan findings. However, we cannot rule out the possibility that these abnormalities were related to imatinib treatment. Considering the short duration of treatment in the COUNTERCOVID study as compared to the median time to development of interstitial lung disease as adverse event, development of imatinib-related interstitial lung disease in COVID-19 is unlikely.”

Reviewer #2 (Remarks to the Author):

Nature comms Duijvelaar E et al. 2023

This manuscript is a very well written and crafted manuscript. The authors did longitudinal proteomic profiling of plasma from 318 COVID19 patients to investigate the effects of critical COVID19. They also assess the effects of imatinib. There was some in vivo hamster experiments to describe these effects in the lungs. Within this research are some really interesting findings and with the deep phenotyping, longitudinal sampling and treatment experiments the project should provide a wealth of new information that can form the basis of clinical based decisions in the management of COVID-19 and other respiratory viral infections. However, my main criticism is not at the experimental design, execution or data analysis. This was all of a pretty high standard. I think on occasion however, the authors would be better off not stating as facts certain assumptions that are implied by the data. The two main examples are the use of the hamster model which the RNAseq data indicates similarities between the human plasma proteome and the hamster lung transcriptome. The conclusion the authors draw is this proves thus that the differential expression shown in human plasma is derived from the pulmonary compartment. This would seem to be a reasonable suggestion, but you are looking at correlation at best. It does not prove that all these changes are derived from lung. For a start, the authors didn't look at any other hamster tissue transcriptomes and looking at some of the proteins these are clearly derived from other tissues such as liver but found in the circulatory system e.g. ApoD (which you demonstrate is upregulated on treatment with Imatinib). Secondly, and I feel the association here is a bit stronger, but I'm not convinced that your data clearly proves that you have alveolar-capillary barrier disruption which is then reversed by Imatinib. I think you have grounds to state that the observed molecular changes are indicative of alveolar-capillary barrier disruption but I'm not convinced you have demonstrated absolute proof. It may feel pendantic the difference, but I think it is important that we recognise the hierarchy of evidence here.

I would like to see a revised title and certain parts of the text where these fundamental absolutist statements are watered down. I personally don't think it will harm the paper because it is a well executed project.

RE: We are grateful for the time and effort put into reviewing this manuscript, and the valuable suggestions and feedback from the reviewer.

In light of the reviewer's comments, we have revised the manuscript so that conclusions are tempered and more nuanced. For example, we now state that external RNA-sequencing analysis was performed to 'contextualize' rather than 'validate' the findings derived from the proteomics analysis (Line 48, 632, 845). The hamster RNAseq findings are furthermore reported as a concurrent or comparative observation, rather than a validation of the human plasma proteome (Line 339, 410). Instead of stating that the plasma proteome reveals alveolar-capillary barrier disruption, we now state throughout the manuscript that 'biomarkers of alveolar-capillary barrier disruption are reflected in plasma (Title and Line 51, 414). The title has been revised to "Longitudinal plasma proteomics reveals biomarkers of alveolar-capillary barrier disruption in critically ill COVID-19 patients". Specifically, for the comparison of our data with hamster RNAseq data, we have removed comments that infer causality in relation to the human proteome. We acknowledge that some of the proteins affected by imatinib treatment are likely not derived from the pulmonary compartment. Following up on the suggestion of reviewer #2 (and that of reviewer

#3), we added findings from an external cellular deconvolution analysis³⁸. This external study applied single-cell and single-nucleus RNA sequencing (sc/snRNA-Seq) across various cell types within tissue samples obtained from patients with fatal COVID-19. Using this dataset, we conducted a deconvolution analysis to investigate how genes encoding protein whose abundance was influenced by imatinib treatment and were enriched in our pathway analyses (as observed in our proteomics analysis), were regulated at the RNA level across various cell types within the lungs of patients who experienced critical illness. Some proteins that were refined, specifically those that are related to the alveolar-capillary barrier, were discussed (Line 376). A significant proportion of proteins was not differentially expressed in the lungs of patients with critical COVID-19 (Figure S15). In line with the statement from reviewer #2, this indicates that these proteins were not regulated on RNA level in the pulmonary compartment.

Some minor issues

Line 101 – aberrant protein expression levels?? Figure S1 doesn't really tell you anything about that?

RE: Flagging of the samples indicated that there was a technical issue while running the assay, and did not pass the quality checks. These flagged samples showed aberrant protein abundance levels. Because the accuracy of the flagged samples could not be guaranteed, they were omitted from further analyses. ("Figure S1") refers to the flowchart in which it is shown that 55 samples were removed after quality control. Nevertheless, because a reference to the quality control was already made at the beginning of the sentence, this reference can be considered redundant and was omitted (Line 122). To make clear that 55 samples were removed after quality control, in Figure S1, "Removed after quality control" was changed to "55 Samples removed after quality control".

Lune 103 – randomly assigned – how? Were you able to maintain age and sex match?

RE: Patient characteristics are stratified by treatment allocation in table S5 in the supplementary appendix. To include more information regarding randomisation, the following text about the randomisation process was added to the methods section: "Randomisation was done with the Castor Electronic Data Capturing System (Castor EDC; Amsterdam, Netherlands). Randomisation was stratified by hospital site using variable block sizes of two, four, or six patients. All patients, health care providers and study investigators were blinded to treatment allocation." For the current proteomics analysis, there were no differences in terms of age (mean placebo 64.3, SD=12.1; mean imatinib 64.4, SD = 12.5, p=0.906) and sex (placebo 66.0%, imatinib 73.5%, p=0.187) between the two treatment groups (Table S5).

Line 156 – "clinical deterioration is reflected by temporal changes in the plasma proteome.

RE: Maybe we misread the comment, but we do not fully grasp the reviewers' comment here. We noticed that a full stop was missing at the end of the sentence. This was added.

Line 187- I think 4a is showing a subset of these proteins. I think that needs explaining in the text.

RE: In brackets: "Figure 4A" was changed to "the 50 most statistically significant proteins are shown in figure 4A". This was also mentioned in the legend of figure 4: "The 50 proteins with the highest significance levels are shown and ordered from high to low log(hazard ratio).

Line 234. – the Referred to Figure S10 doesn't really show anything

RE: Figure S10 shows that at baseline there are no differences between protein abundance in patients allocated to either imatinib or placebo. The most important message is that randomization was well balanced and that there were no baseline differences between the two treatment groups with regards to plasma protein abundance composition. While technically, this could be read from Supplementary table 6, this is much more reader friendly.

Line 238 – 5A should be 5B

RE: We thank the reviewing for noticing this, and assume that the reviewer referred to 4B instead of 5A. "4B" was changed to "5B"

Figure 5 B what is n?

RE: The results which are displayed in Figure 4 are conducted using plasma samples collected at hospital admission. All the other analyses, including those presented in figure 5, were conducted using plasma samples from 156 patients treated with placebo and 162 treated with imatinib, totalling 318 patients. We added to the statistical analysis section: "All statistical analyses were conducted on plasma samples collected from all 318 patients, unless stated otherwise." "(n=249)" was additionally added to the two results sections related to the association with development of critical illness since these were analysed using plasma baseline samples.

Role of comorbidities should be underplayed as they are significantly different in your cohort

RE: Comorbid diabetes mellitus, cardiovascular disease (including atrial fibrillation) and BMI (as a surrogate for obesity) were significantly different between patients with a non-critical versus critical illness (Figure 1). We believe that these comorbidities should not be underplayed since they are associated with critical illness. When patients with critical illness are compared with patients with a non-critical illness, this should be accounted for. For that reason, these factors were included as covariates in the linear mixed models (line 767). Additionally, we accounted for age, sex, hospital site and treatment allocation.

For the evaluation of the effect of imatinib, these comorbidities should not be included in the linear mixed models. The main reason for this, is that the effect of imatinib was calculated within each patient rather than between different patients. These are paired observations; the protein abundance at day 0 was compared to the protein abundance at day 3. Each patient serves as their own comparator. To account for related observations within patients, a random intercept was entered for each patient. Moreover, patients were randomised for treatment allocation, and none of the proteins had a different abundance between the treatment groups at baseline.

Line 256 – surplus comma

RE: The surplus comma was omitted.

I think the data interrogation of the hamster data does not necessarily prove that this is a lung-isolated mechanism in humans.

RE: We acknowledge that presenting both results does not infer causality and therefore tempered our conclusion; no longer stating that the hamster RNA seq data validates clinical findings (Line 48, 632, 845). Moreover, we discuss the limitations of comparing proteomics analysis in peripheral blood with RNA sequencing data on lung tissue (Line 388, 637). The findings from our analysis on the effects of imatinib were contextualized using external sc/snRNA sequencing data obtained from tissue samples obtained from patients with fatal COVID-19 (Line 374). These findings illustrate how proteins affected in the plasma proteome are up or downregulated at the RNA level in the pulmonary compartment. In line with the statement from the reviewer, and mentioned earlier, a significant proportion of proteins are not regulated at RNA level in the pulmonary compartment.

Line 476-486 What volume of plasma is used per assay?

RE: Approximately 55 μ L plasma was used for proteomic analyses, stored in 500 μ L aliquots. This information was added to the methods.

Reviewer #3 (Remarks to the Author):

In this original research manuscript, the authors analyzed serum proteomic data from 385 patients from a clinical trial in which imatinib had been tested as a therapy for COVID-19. They found that most proteins increased at enrollment among subjects who developed critical illness proteins remained increased levels at day 3, and that certain proteins followed disease-outcome-specific trajectories. LASSO-based regression identified a group of proteins that predicted the subsequent development of critical illness. Surprisingly, there was only a modest effect of imatinib on plasma proteins. Overall, the study was well-designed and the analyses follow reasonable and appropriate statistical approaches, although an external validation cohort would have been preferable.

Recognizing these strengths, the size/scope of the study and the challenge of such a study during the pandemic, it is a bit less clear what specific new insights into SARS-CoV-2 pathobiology or severe critical illness the results add beyond extensive literature on both of these topics – specifically, the principal assertion that = disruption of the alveolar-capillary barrier is related to disease severity/progression is well-aligned with the current paradigm of acute lung injury/ARDS, albeit of somewhat limited novelty.

RE: We appreciate the time and effort that the reviewer put into reviewing the manuscript, and thank the reviewer for the thorough read. Although we acknowledge that alveolar-capillary barrier disruption is a known phenomenon in acute lung injury/ARDS, an important caveat in clinical medicine is that there are currently no useful biomarkers to measure alveolar-capillary barrier disruption in peripheral blood. This study provides novel information for several reasons. First, large scale proteomic analysis allowed us to investigate a vast larger number of blood derived biomarkers, and yielded previously unidentified blood derived markers of alveolar-capillary disruption. This could allow future studies (or possibly even in the clinic) to measure markers of alveolar capillary disruption. Second, our study comprises a comprehensive analysis of a very large number of proteins measured in peripheral blood in a population of patients with a relatively homogenous clinical status (i.e. almost all patients (92%) at time of baseline blood collection, were admitted to the ward and received supplemental oxygen through a nasal cannula or mask, corresponding to category 3 on the WHO ordinal scale). This design is necessary to understand which patients develop an ARDS phenotype, and through which mechanisms. Last, this study is unique in that it has performed proteomic analysis during a randomized intervention. No study has previously evaluated the effect of imatinib, or any other drug on the wide-scale plasma proteome in COVID-19.

Major comments:

Many of the violin/box plots are missing statistical annotation.

RE: Statistical annotations were added to the violin plots in Figure 2, and for the boxplots in Figures 3 and 5.

The use of bulk lung tissue RNA-seq data from Syrian hamsters infected with SARS-CoV-2 really provides little added mechanistic insight as there seemed to be modest concordance with imatinib-interacting proteins. If a “cellular attribution”-type deconvolution analysis was sought, it could be performed using single-cell data from

SARS-CoV-2 tissue atlases (for example PMID 33915568, 33915569).

RE: We would like to thank the reviewer for the suggestion and for providing these references. We acknowledge that both the COVID-19 tissue atlas and our plasma proteome study are parallel observations and do not imply causality. Specifically, we think that the study performed by Delorey et al.³⁸ could indeed be used to contextualize our findings with regards to the effect of imatinib on the human plasma proteome. The study performed by Delorey et al, applied single-cell and single-nucleus RNA sequencing (sc/snRNA-Seq) across various cell types within tissue samples obtained from patients with fatal COVID-19. Using this dataset, we conducted a deconvolution analysis to investigate how genes encoding proteins, whose abundance was influenced by imatinib treatment and were enriched in our pathway analyses (as observed in our proteomics analysis), were regulated at the RNA level across various cell types within the lungs of patients experiencing critical illness. Concurrently, we acknowledge that comparing RNAseq analysis on pulmonary tissue with plasma proteomics has its limitations (Line 388, 637).

The study would be considerably strengthened by external cohort validation of the predictive protein signature.

RE: Following up on the suggestions of the reviewer, we incorporated external cohort validation using two study cohorts that provided open access to their data (Figure S10). We selected two independent proteomics studies that employed Somascan assays. The first study enrolled 54 patients with end-stage kidney disease (ESKD) who contracted COVID-19 and were admitted to the Hammersmith Hospital (HH) in London, United Kingdom¹. To the best of our knowledge, this was the only study that included all nine proteins and had follow-up data on critical illness available. In comparison with our internal VUMC validation cohort (AUC=0.813) the model exhibited comparable accuracy in predicting critical illness development in the HH study cohort (AUC = 0.801). We additionally tested the performance in a second external dataset, derived from a study conducted at the Massachusetts General Hospital (MGH) in Boston, United States of America, and assessed protein abundance in 308 patients with COVID-19 upon hospital admission². However, in the MGH dataset only five out of nine proteins from our signature set were available. External validation of this 5-protein signature in the MGH yielded an even higher accuracy (AUC = 0.838) than in our internal validation cohort. When reducing our set of nine proteins to the five proteins available in the MGH cohort, we achieved an accuracy of AUC = 0.802 in the VUMC internal validation cohort. These findings indicate that our protein signature set has similar performance in external study cohorts. Differences in performance may, in part, be attributed to variations in the proportion of patients who developed critical illness. In the CounterCOVID trial, this proportion was 21.7% (69 out of 318 patients), which more closely resembles the proportion in the HH cohort (16.6%, or 51 out of 301 patients) than that of the MGH cohort (35.6%, or 109 out of 306 patients). External validation was only feasible for the protein only model due to the restricted availability of clinical variables that we selected for our models.

Cohort	AUC
VUMC internal validation (9-protein model)	0.813
HH external validation (9-protein model)	0.801

VUMC internal validation (5-protein model)	0.802
MGH external validation (5-protein model)	0.838

Minor comment:

Throughout the text, the terms upregulated/downregulated and or differentially-expressed are used to describe proteins with different abundance – it is a minor point, but these differences likely do not result from a regulated process or differences in expression, but rather leak into circulation from injured/dying cells – the text should be clarified to more precisely note differential abundance or increased/decreased levels.

RE: We thank the reviewer for mentioning this. Expression and up/downregulation should indeed exclusively be used for the overall process of a gene being turned into a protein. This was changed throughout the manuscript.

The discussion is a bit lengthy and speculative, this could be shortened.

RE: The paragraphs starting with “We identified 538 proteins...” and “in the study performed ...” were almost completely omitted and key sentences were incorporated in other paragraphs. Furthermore, the first paragraph and some other sentences of the discussion were shortened. This resulted in a reduction of 495 words. However, following up on the suggestions of reviewer #1 we added a new paragraph related to the differential effects of other break point cluster (Bcr) Abelson (Abl) tyrosine-kinase inhibitors and rare reports of development of imatinib related interstitial lung disease. In addition, following up on the suggestions of reviewer #4, conformity between methodological proteomics platforms is discussed.

Reviewer #4 (Remarks to the Author):

In this paper the authors report results from a proteome analyses using aptamer technique (6385 plasma proteins measured) in blood from patients in an RCT to test how imatinibe affects outcome in hospitalized patients with Covid 19. The proteome is assessed at hospital admission and 3 days later.

The paper has several strengths including the relatively large sample size, serial sampling and the randomized intervention with a tyrosine kinase inhibitor. Thus, the study does not only inform on proteome perturbations in Covid 19 and critical illness but also how imatinibe affects the proteome.

I find the analyses well performed and the paper well presented and written. There is transparent reporting of QC procedures and results.

A weakness that is inherent to most studies using explorative proteomics is the difficulties in interpretation of the results. A number of positive signals are reported that suggest regulation of certain networks, on the contrary, critical proteins that are not regulated are difficult to interpret and sometimes underreported. There is also a lack of conformity between different methodological platforms used and aptamer based studies are not always supported by antibody techniques.

RE: We would like to thank the reviewer for the time invested in thoroughly reviewing the manuscript.

Questions and comments:

It is unclear to me when plasma samples were retrieved in relation to Imatinib. Were plasma samples retrieved 3 days after starting Imatinib or three days after inclusion? This is crucial for interpretation of the data and should be clarified.

RE: Baseline plasma samples were collected within 24 hours of hospital admission and before the start of imatinib/placebo treatment. We have added a statement to the results section that baseline samples were collected before the start of imatinib or placebo (Line 312). Samples were not immediately collected upon hospital admission because informed consent, randomization and dispensing of the study drugs required some time. Because imatinib was more effective upon earlier administration in animal models³⁹, inclusion and first study drug administration were usually only several hours apart. Follow-up samples were collected at study day 3, which was approximately 72 hours after the start of imatinib/placebo treatment. To make this clearer in the method section, “at Day 3 of the study” was changed to: “3 days after first study drug administration (Day 3)” (Line 709).

I find it strange that the “clinical + protein model” performs worse than the “protein alone model” when tested in the validation cohort. Giving the names (and number) of the proteins included in the “clinical+protein model” versus the “protein alone model” could clarify and show that other proteins were selected.

RE: The names (Gene IDs and/or clinical variables) of the variables selected for each model can be found in Figure 4E. Additionally, to ensure that it is clear to readers which variables were selected, we have added their names to the text on lines 258-270. The performance in the protein + clinical model was only superior to the protein only model in the validation cohort. Given that the protein only model was inferior in the training cohort, indicates that the clinical

variables were less applicable to unseen cohorts. Moreover, it suggests that proteins better capture biological processes than the clinical phenotype. In addition, it indicates that protein signatures perform more stable between cohorts than clinical data. Other studies hint towards the same conclusion in which the proteome is more accurate compared to clinical variables in the recognition of disease severity^{40,41}. This may be partly explained by local differences in clinical management that influence clinical parameters independent of underlying biology.

Which clinical variables were used for training (same as in Table 1?) and what variables were selected for prediction models?

RE: All variables, except for variables belonging to drug administrations and clinical course, displayed in table 1 were entered for the LASSO regression models. Additionally, all measured proteins, several comorbidities, smoking pack years, vital parameters, laboratory assessments and QTc interval were included. We added a column to the data dictionary belonging to Supplementary data file 1 indicating which variables were included in the LASSO regression models. Additionally, supporting information regarding the selected variables have been incorporated in the manuscript in Line 265: “In the clinical-only model the predictors included: thrombocyte count, oxygen saturation measured by pulse oximetry/fraction of inspired oxygen (SpO₂/FiO₂), comorbid diabetes mellitus, high-sensitive cardiac troponin T (hs-cTnT), N-terminal pro-B-type natriuretic peptide (NT-proBNP), age and SpO₂. In the combined protein and clinical model, the selected predictors consisted of: SpO₂/FiO₂, DR1, RBFOX1 and PFDN4 (Figure 4E).”

Include a statement that “aptamer based proteomics was used” in the last paragraph of the introduction. This is central information for the reader.

RE: The following sentence was added to the last paragraph of the introduction: “The protein abundance of 6385 unique proteins or protein complexes was assessed using the aptamer-based Somascan platform.”

Was there a rationale for selecting aptamer based techniques (somascan) in favor of antibody techniques (OLINK)? Please comment in the text of the likelihood of being able to reproduce the results using other platforms? (see Katz et al., Sci. Adv. 2022) This could be especially relevant for the prediction model based on 9 proteins.

RE: The primary reason for selecting the Somascan assay was its ability to provide comprehensive coverage of proteins due to its high number of protein targets. We acknowledge that different platforms have their unique strengths and weaknesses. Moreover, the correlation between the measurements from Somascan and Olink show a wide range, indicating that the inter-platform reproducibility varies strongly for each protein target⁴². We therefore believe that, when applying high throughput proteomics, robust conclusions can only be derived from pathway analysis (often including a large number of proteins which offer internal validation), rather than from the observations of individual proteins.

We encountered limited information regarding the inter-platform performance of our panel of nine proteins. This is largely due to the fact that previous studies utilizing the Olink platform incorporated only two of the nine proteins that we identified for our protein-based prediction model: IL6 and

MEPE^{2,40,41,43-46}. According to the study from Katz et al., the Pearson correlation coefficient (ρ) for MEPE and IL6 are reported as 0.12 and 0.49 respectively⁴². These values do not suggest a high level of inter-platform reproducibility, however information on the other seven proteins is lacking. The good performance in external cohort validation (Line 281) indicates high reproducibility in two other independent study cohorts that applied the Somascan assays. However, its reproducibility using the Olink platform cannot be formally tested due to the restricted number of available protein targets.

In a previously published study, the protein plasma concentration of biomarkers were measured in plasma derived from the same patients and time points⁶. Plasma concentrations were measured using Luminex (R&D Systems Inc., Minneapolis, United States) multiplex assays. To assess conformity between the two platforms, we applied Pearson correlation on nine overlapping proteins. Pearson correlation coefficients indicate that some proteins exhibited moderate to strong correlations (IL-6 $\rho=0.80$, IL-8 $\rho=0.51$ and RAGE $\rho=0.74$), while others showed very weak or even negative correlations (IL-10 $\rho=0.14$, and IL-2 $\rho=-0.05$).

To convey these messages to the readers, we have incorporated the strength and weaknesses of both platforms (Line 612), the inter-platform reproducibility (Line 618) and correlations with the measurements from our Luminex assays (Line 621) into the discussion section.

The sentence is incomplete: Page 3, p 83-83: “Identification of pathways that contribute to critical illness may highlight novel biomarkers that could be used risk for risk stratification and provide targets for the treatment of respiratory failure in 85 COVID-19 and acute respiratory distress syndrome (ARDS).”

RE: The word “risk” was unintentionally written twice. This was corrected. After this correction, we believe that the sentence is complete. It presents the idea that “identification of novel pathways” (subject) could be used for 1. risk stratification and 2. provide targets for the treatment of respiratory failure in COVID-19 and ARDS (predicates).

Missing data on BMI was imputed. Where no other proteindata imputed?

RE: As stated in line 712 of the methods section, in cases where a plasma sample from the third day was unavailable, the sample from the second day was utilized as a substitute. Apart from this, and the imputation of missing data on BMI with the median BMI value, no further imputation was performed.

References related to the point-by-point response

1. Gisby JS, Buang NB, Papadaki A, Clarke CL, Malik TH, Medjeral-Thomas N, et al. Multi-omics identify falling LRRRC15 as a COVID-19 severity marker and persistent pro-thrombotic signals in convalescence. *Nat Commun.* 2022 Dec;13(1):7775.
2. Filbin MR, Mehta A, Schneider AM, Kays KR, Guess JR, Gentili M, et al. Longitudinal proteomic analysis of severe COVID-19 reveals survival-associated signatures, tissue-specific cell death, and cell-cell interactions. *Cell reports Med.* 2021 May;2(5):100287.
3. Leisman DE, Mehta A, Thompson BT, Charland NC, Gonye ALK, Gushterova I, et al. Alveolar, Endothelial, and Organ Injury Marker Dynamics in Severe COVID-19. *Am J Respir Crit Care Med.* 2021 Dec;
4. Aman J, Duijvelaar E, Botros L, Kianzad A, Schippers JR, Smeele PJ, et al. Imatinib in patients with severe COVID-19: a randomised, double-blind, placebo-controlled, clinical trial. *Lancet Respir Med.* 2021;9(9):957–68.
5. Duijvelaar E, Schippers JR, Smeele PJ, de Raaf MA, Vanhove ALEM, Blok SG, et al. Long-term clinical outcomes of COVID-19 patients treated with imatinib. *The Lancet Respiratory Medicine.* 2022.
6. de Brabander J, Duijvelaar E, Schippers JR, Smeele PJ, Peters-Sengers H, Duitman JW, et al. Immunomodulation and endothelial barrier protection mediate the association between oral imatinib and mortality in hospitalised COVID-19 patients. *Eur Respir J.* 2022 Jul;
7. Baalbaki N, Duijvelaar E, Said MM, Schippers J, Bet PM, Twisk J, et al. Pharmacokinetics and pharmacodynamics of imatinib for optimal drug repurposing from cancer to COVID-19. *Eur J Pharm Sci Off J Eur Fed Pharm Sci.* 2023 May;184:106418.
8. Bartelink IH, Bet PM, Widmer N, Guidi M, Duijvelaar E, Grob B, et al. Elevated acute phase proteins affect pharmacokinetics in COVID-19 trials: Lessons from the CounterCOVID - imatinib study. *CPT pharmacometrics Syst Pharmacol.* 2021 Dec;10(12):1497–511.
9. Botros L, Pronk MCA, Juschten J, Liddle J, Morsing SKH, van Buul JD, et al. Bosutinib prevents vascular leakage by reducing focal adhesion turnover and reinforcing junctional integrity. *J Cell Sci.* 2020 May;133(9).
10. Aman J, van Bezu J, Damanafshan A, Huveneers S, Eringa EC, Vogel SM, et al. Effective treatment of edema and endothelial barrier dysfunction with imatinib. *Circulation.* 2012 Dec;126(23):2728–38.
11. Amado-Azevedo J, van Stalborch A-MD, Valent ET, Nawaz K, van Bezu J, Eringa EC, et al. Depletion of Arg/Abl2 improves endothelial cell adhesion and prevents vascular leak during inflammation. *Angiogenesis.* 2021 Aug;24(3):677–93.
12. El-Agamy DS. Nilotinib attenuates endothelial dysfunction and liver damage in high-cholesterol-fed rabbits. *Hum Exp Toxicol [Internet].* 2017;36(11):1131–45. Available from: <https://pubmed.ncbi.nlm.nih.gov/27941169/>
13. Chan M, Vijay S, McNevin J, McElrath MJ, Holland EC, Gujral TS. Machine learning identifies molecular regulators and therapeutics for targeting SARS-CoV2-induced cytokine release. *Mol Syst Biol.* 2021 Sep;17(9):e10426.
14. Liu Z, Chen S, Zhang X, Liu F, Yang K, Du G, et al. Dasatinib protects against acute respiratory distress syndrome via Nrf2-regulated M2 macrophages polarization. *Drug Dev Res.* 2021 Dec;82(8):1247–57.
15. Han Y, Duan X, Yang L, Nilsson-Payant BE, Wang P, Duan F, et al. Identification of SARS-CoV-2 inhibitors using lung and colonic organoids. *Nature.* 2021 Jan;589(7841):270–5.
16. Rizzo AN, Aman J, van Nieuw Amerongen GP, Dudek SM. Targeting Abl kinases to regulate vascular leak during sepsis and acute respiratory distress syndrome. *Arterioscler Thromb Vasc Biol.* 2015 May;35(5):1071–9.
17. Letsiou E, Rizzo AN, Sammani S, Naureckas P, Jacobson JR, Garcia JGN, et al.

- Differential and opposing effects of imatinib on {LPS}- and ventilator-induced lung injury. *Am J Physiol Lung Cell Mol Physiol*. 2015 Feb;308(3):L259--269.
18. Zaki OS, Safar MM, Ain-Shoka AA, Rashed LA. A Novel Role of a Chemotherapeutic Agent in a Rat Model of Endotoxemia: Modulation of the STAT-3 Signaling Pathway. *Inflammation*. 2018 Feb;41(1):20–32.
 19. Xin Y, Cereda M, Yehya N, Humayun S, Delvecchio P, Thompson JM, et al. Imatinib alleviates lung injury and prolongs survival in ventilated rats. *Am J Physiol Lung Cell Mol Physiol*. 2022 Jun;322(6):L866–72.
 20. Li Z, Peng M, Chen P, Liu C, Hu A, Zhang Y, et al. Imatinib and methazolamide ameliorate COVID-19-induced metabolic complications via elevating ACE2 enzymatic activity and inhibiting viral entry. *Cell Metab*. 2022 Mar;34(3):424-440.e7.
 21. Xia L, Yuan L-Z, Hu Y-H, Liu J-Y, Hu G-S, Qi R-Y, et al. A SARS-CoV-2-specific CAR-T-cell model identifies felodipine, fasudil, imatinib, and caspofungin as potential treatments for lethal COVID-19. *Cell Mol Immunol*. 2023 Mar;1–14.
 22. Touret F, Driouich J-S, Cochin M, Petit PR, Gilles M, Barthélémy K, et al. Preclinical evaluation of Imatinib does not support its use as an antiviral drug against SARS-CoV-2. *Antiviral Res*. 2021 Sep;193:105137.
 23. Macfarlane JG, Dorward DA, Ruchaud-Sparagano M-H, Scott J, Lucas CD, Rossi AG, et al. Src kinase inhibition with dasatinib impairs neutrophil function and clearance of *Escherichia coli* infection in a murine model of acute lung injury. *J Inflamm (Lond)*. 2020 Oct;17(1):34.
 24. Mohty M, Blaise D, Olive D, Gaugler B. Imatinib: the narrow line between immune tolerance and activation. *Trends Mol Med*. 2005 Sep;11(9):397–402.
 25. Ciarcia R, Vitiello MT, Galdiero M, Pacilio C, Iovane V, d'Angelo D, et al. Imatinib treatment inhibit IL-6, IL-8, NF-KB and AP-1 production and modulate intracellular calcium in CML patients. *J Cell Physiol*. 2012 Jun;227(6):2798–803.
 26. Weatherald J, Bondeelle L, MC C, Guignabert C, Savale L, Jaïs X, et al. Pulmonary complications of Bcr-Abl tyrosine kinase inhibitors. *Eur Respir J [Internet]*. 2020;56(4). Available from: <https://pubmed.ncbi.nlm.nih.gov/32527740/>
 27. JJ M, Deininger M. Tyrosine Kinase Inhibitor-Associated Cardiovascular Toxicity in Chronic Myeloid Leukemia. *J Clin Oncol Off J Am Soc Clin Oncol [Internet]*. 2015;33(35):4210–8. Available from: <https://pubmed.ncbi.nlm.nih.gov/26371140/>
 28. Gover-Proaktor A, Granot G, Pasmanik-Chor M, Pasvolsky O, Shapira S, Raz O, et al. Bosutinib, dasatinib, imatinib, nilotinib, and ponatinib differentially affect the vascular molecular pathways and functionality of human endothelial cells. *Leuk Lymphoma*. 2019 Jan;60(1):189–99.
 29. Haguët H, Bouvy C, AS D, Modaffari E, Wannez A, Sonveaux P, et al. The Risk of Arterial Thrombosis in Patients With Chronic Myeloid Leukemia Treated With Second and Third Generation BCR-ABL Tyrosine Kinase Inhibitors May Be Explained by Their Impact on Endothelial Cells: An In-Vitro Study. *Front Pharmacol [Internet]*. 2020;11:1007. Available from: <https://pubmed.ncbi.nlm.nih.gov/32719607/>
 30. Guignabert C, Phan C, Seferian A, Huertas A, Tu L, Thuillet R, et al. Dasatinib induces lung vascular toxicity and predisposes to pulmonary hypertension. *J Clin Invest*. 2016 Sep;126(9):3207–18.
 31. Paez-Mayorga J, AL C, Kotla S, Tao Y, RJ A, ED H, et al. Ponatinib Activates an Inflammatory Response in Endothelial Cells via ERK5 SUMOylation. *Front Cardiovasc Med [Internet]*. 2018;5:125. Available from: <https://pubmed.ncbi.nlm.nih.gov/30238007/>
 32. Valent P, Hadzijušufovic E, GH S, Wolf D, Rea D, P le C. Vascular safety issues in CML patients treated with BCR/ABL1 kinase inhibitors. *Blood [Internet]*. 2015;125(6):901–6. Available from: <https://pubmed.ncbi.nlm.nih.gov/25525119/>
 33. Duijvelaar E, Vanhove A, Schippers JR, Smeele PJ, de Man FS, Pinto Y, et al. Cardiac safety of imatinib for the treatment of Covid-19: a secondary analysis of a randomised, double blind, placebo-controlled trial. *J Cardiovasc Pharmacol*. 2022 Aug;
 34. Ohnishi K, Sakai F, Kudoh S, Ohno R. Twenty-seven cases of drug-induced interstitial lung disease associated with imatinib mesylate. Vol. 20, *Leukemia*. England; 2006. p.

- 1162–4.
35. Isshiki I, Yamaguchi K, Okamoto S. Interstitial pneumonitis during imatinib therapy. *Br J Haematol*. 2004 May;125(4):420.
 36. Zhang P, Huang J, Jin F, Pan J, Ouyang G. Imatinib-induced irreversible interstitial lung disease: A case report. *Medicine (Baltimore)*. 2019 Feb;98(8):e14402.
 37. Rosado MF, Donna E, Ahn YS. Challenging problems in advanced malignancy: Case 3. Imatinib mesylate-induced interstitial pneumonitis. *J Clin Oncol Off J Am Soc Clin Oncol*. 2003 Aug;21(16):3171–3.
 38. Delorey TM, Ziegler CGK, Heimberg G, Normand R, Yang Y, Segerstolpe Å, et al. COVID-19 tissue atlases reveal SARS-CoV-2 pathology and cellular targets. *Nature*. 2021 Jul;595(7865):107–13.
 39. Kim IK, Rhee CK, Yeo CD, Kang HH, Lee DG, Lee SH, et al. Effect of tyrosine kinase inhibitors, imatinib and nilotinib, in murine lipopolysaccharide-induced acute lung injury during neutropenia recovery. *Crit Care*. 2013 Jun;17(3):R114.
 40. Al-Nesf MAY, Abdesselem HB, Bensmail I, Ibrahim S, Saeed WAH, Mohammed SSI, et al. Prognostic tools and candidate drugs based on plasma proteomics of patients with severe COVID-19 complications. *Nat Commun*. 2022 Feb;13(1):946.
 41. Gisby J, Clarke CL, Medjeral-Thomas N, Malik TH, Papadaki A, Mortimer PM, et al. Longitudinal proteomic profiling of dialysis patients with COVID-19 reveals markers of severity and predictors of death. *Elife [Internet]*. 2021 Mar 11;10. Available from: <https://elifesciences.org/articles/64827>
 42. Katz DH, Robbins JM, Deng S, Tahir UA, Bick AG, Pampana A, et al. Proteomic profiling platforms head to head: Leveraging genetics and clinical traits to compare aptamer- and antibody-based methods. *Sci Adv*. 2022 Aug;8(33):eabm5164.
 43. Zhong W, Altay O, Arif M, Edfors F, Doganay L, Mardinoglu A, et al. Next generation plasma proteome profiling of COVID-19 patients with mild to moderate symptoms. *eBioMedicine [Internet]*. 2021 Dec;74:103723. Available from: <https://linkinghub.elsevier.com/retrieve/pii/S235239642100517X>
 44. Byeon SK, Madugundu AK, Garapati K, Ramarajan MG, Saraswat M, Kumar-M P, et al. Development of a multiomics model for identification of predictive biomarkers for COVID-19 severity: a retrospective cohort study. *Lancet Digit Heal*. 2022 Sep;4(9):e632–45.
 45. Feyaerts D, Hédou J, Gillard J, Chen H, Tsai ES, Peterson LS, et al. Integrated plasma proteomic and single-cell immune signaling network signatures demarcate mild, moderate, and severe COVID-19. *Cell reports Med*. 2022 Jul;3(7):100680.
 46. Keur N, Saridaki M, Ricaño-Ponce I, Netea MG, Giamarellos-Bourboulis EJ, Kumar V. Analysis of inflammatory protein profiles in the circulation of COVID-19 patients identifies patients with severe disease phenotypes. *Respir Med*. 2023 Oct;217:107331.

REVIEWERS' COMMENTS

Reviewer #1 (Remarks to the Author):

I have reviewed the revision. The authors successfully responded to my comments/suggestions, and thus, for my point of view, the manuscript can be accepted for publication in its current form.

Reviewer #2 (Remarks to the Author):

I acknowledge the considerable changes made to the document and I am happy with all the responses. I particularly liked the toning down of conclusions which do not lessen the impact of the paper and I think the addition of the external cellular deconvolution analysis strengthens the paper. Overall, I think this is a much stronger paper ready for publication

One minor point is in respect to my original minor point I suggested 5A should be 5B in what was line 238. The Authors rightly noticed I actually meant 4B and that this should be changed to 5B. However, the authors have changed this to 5A which I think is incorrect, and possibly influenced by my initial error. Please change to 5B. (its now line 317)

Reviewer #4 (Remarks to the Author):

I find that the authors have responded to the questions raised. The paper has improved especially with the external validation.

Reviewer #5 (Remarks to the Author):

The authors have addressed the comments from the previous reviewer. In particular the validation of the results in the two external data set is an important addition to the manuscript.

Still, there are some minor points that the authors should address. For example, in Figure 2B there are instances where the same protein name is shown several times in the Volcano plot (e.g. FN1). Why is this?

Second, it would be helpful if the authors could state how many of the SOMAmer-protein bound complexes have been characterized for the proteins that were included in the top-9 panel (RNA binding protein fox-1 homolog 2 (RBFOX2), splicing factor 45 (RBM17), protein Dr1 (DR1), Prefoldin subunit 4 (PFDN4), translocon-associated protein subunit beta (SSR2), ataxin-2-binding protein 1) and for the proteins that were associated with critical illness and reversed by Imatinib (repulsive guidance molecule a (RGMa), thrombospondin-2, IL-6, biglycan and urokinase-type plasminogen activator (PLAU)). Are the chemically modified DNA sequences that bind to the proteins of interest, specific for their target protein? In general, The SOMAscan assay is designed as a discovery platform and measures relative protein concentrations using external controls. Without internal controls and standard curves, it remains unclear how specific the measurements are and which measurements are within the linear dynamic range. It would be helpful if the authors could comment on 1) the level of validation that has been performed for the aptamers that recognize these proteins, 2) how many aptamers identified each of these proteins and 3) how consistent the different aptamers were for these proteins if there was more than one. Another orthogonal validation of the changes in protein abundances for these proteins (or other proteins for that matter) using for example ELISA or other specific methods would substantially increase the confidence in the reported results.

Point-by-point response on editorial comments:

Reviewer #1 (Remarks to the Author):

I have reviewed the revision. The authors successfully responded to my comments/suggestions, and thus, for my point of view, the manuscript can be accepted for publication in its current form.

RE: We would like to express our gratitude to reviewer #1 for recognizing the completeness of the changes made to the manuscript

Reviewer #2 (Remarks to the Author):

I acknowledge the considerable changes made to the document and I am happy with all the responses. I particularly liked the toning down of conclusions which do not lessen the impact of the paper and I think the addition of the external cellular deconvolution analysis strengthens the paper. Overall, I think this is a much stronger paper ready for publication

One minor point is in respect to my original minor point I suggested 5A should be 5B in what was line 238. The Authors rightly noticed I actually meant 4B and that this should be changed to 5B. However, the authors have changed this to 5A which I think is incorrect, and possibly influenced by my initial error. Please change to 5B. (its now line 317)

RE: We would like to extend our gratitude to reviewer #2 for recognizing the changes made to the manuscript and rightfully noting the incomplete reference. The reference has been corrected to '5b'.

Reviewer #4 (Remarks to the Author):

I find that the authors have responded to the questions raised. The paper has improved especially with the external validation.

RE: We would like to extend our gratitude to Reviewer #4 for recognizing the changes made to the manuscript.

Reviewer #5 Comments:

The authors have addressed the comments from the previous reviewer. In particular the validation of the results in the two external data set is an important addition to the manuscript.

RE: We would like to extend our gratitude to reviewer #5 for taking over the role of reviewer #3. Furthermore, we are pleased that reviewer #5 acknowledges that all the comments raised by reviewer #3 have been appropriately addressed.

Still, there are some minor points that the authors should address. For example, in Figure 2B there are instances where the same protein name is shown several times in the Volcano plot (e.g. FN1). Why is this?

RE: The utilized Somascan library includes 7288 aptamers to target 6385 unique proteins or protein complexes. While the vast majority of proteins are targeted by one aptamer (N=5963), for 632 proteins, multiple aptamers were developed (584 proteins are targeted by 2 aptamers, 44 proteins by 3 aptamers, and 4 proteins by more than 3 aptamers). This is the case, for example, with interleukin-6 (IL6), for which 2 aptamers are available that target different epitopes. Reasons to develop multiple aptamers for a single protein include enhanced specificity, the existence of different isoforms (e.g., FGF8), the existence of different protein conformations (e.g., C3), or the existence of different protein components. In the specific case of FN1, the gene is represented by 4 aptamers in the Somascan library, each targeting different entities: Fibronectin, Fibronectin-1 Fragment 2, Fibronectin Fragment 3, and Fibronectin Fragment 4. A comprehensive list of these genes and their corresponding protein targets can be found in the supplementary Excel file under the "protein content" tab. Researchers interested in exploring the impact of critical illness on each of the FN1 fragments shown in Figure 2B could refer to the Supplementary data file 3 for further insights.

To apprise readers of the aforementioned details, we included a reference to the protein content in the method section (Line 575 in the revised manuscript with track changes). "For some proteins, more than one aptamer is available, to recognize distinct epitopes, isoforms, protein conformations or protein fragments, or a combination of these (detailed in the protein content of the Supplementary data file)."

Second, it would be helpful if the authors could state how many of the SOMAmer-protein bound complexes have been characterized for the proteins that were included in the top-9 panel (RNA binding protein fox-1 homolog 2 (RBFOX2), splicing factor 45 (RBM17), protein Dr1 (DR1), Prefoldin subunit 4 (PFDN4), translocon-associated protein subunit beta (SSR2), ataxin-2-binding protein 1) and for the proteins that were associated with critical illness and reversed by Imatinib (repulsive guidance molecule a (RGMa), thrombospondin-2, IL-6, biglycan and urokinase-type plasminogen activator (PLAU)). Are the chemically modified DNA sequences that bind to the proteins of interest, specific for their target protein? In general, The SOMAScan assay is designed as a discovery platform and measures relative protein concentrations using external controls. Without internal controls and standard curves, it remains unclear how specific the measurements are and which measurements are within the linear dynamic range. It would be helpful if the authors could comment on 1) the level of validation that has been performed for the aptamers that recognizes these proteins, 2) how many aptamers identified each of these proteins and 3) how consistent the different aptamers were for these proteins if there was more than one. Another

orthogonal validation of the changes in protein abundances for these proteins (or other proteins for that matter) using for example ELISA or other specific methods would substantially increase the confidence in the reported results.

RE: 1) Characterization, specificity, and orthogonal method validation for the requested proteins is specified below. For 4 out of 9 (RBM17, NPPA, MEPE and IL6) aptamers of the signature set, specificity was confirmed through either singular or multiple assessments using ELISA, PEA, pQTL, or bead-based immunoassays. For the other 5 aptamers, the SOMAmer reagent was selected using a recombinant construct. Confirmation is currently lacking for these proteins presumably because these were not available in the earlier 5k or 1.3k Somasca libraries¹. RBM17, NPPA, MEPE and IL6 constitute 4 out of 5 of proteins used for external cohort validation in the Filbin cohort (Figure S10; RBFOX2). The fact that the 5-protein signature yielded similar performance, suggests that these are the most important determinants in the prediction of critical illness development.

In a previously published study, the protein plasma concentration of biomarkers were measured in plasma derived from the same patients and time points². Plasma concentrations were measured using Luminex (R&D Systems Inc., Minneapolis, United States) multiplex bead-based immunoassays. We observed a Pearson correlation coefficient of $\rho=0.80$ for IL-6 between the Luminex and Somasca measurements. From the proteins mentioned by the reviewer, IL-6 was the only aptamer for which orthogonal validation could be specified. In response to reviewer #4 we had already included this post hoc comparison with the Luminex assay (Line 478 in the revised manuscript with track changes) and emphasized that “robust conclusions should be derived primarily from pathway analysis, rather than from the observations of individual proteins” (Line 484 in the revised manuscript with track changes).

In addition, as specificity was not confirmed for all aforementioned proteins, we emphasize the need for validation if the nine-protein signature is to be applied in clinical practice. The following sentence has been added to the discussion section: “The specificity of four proteins (RBM17, NPPA, MEPE, and IL6) in the predictive nine-protein signature for critical illness development, was confirmed through either singular or multiple assessments, including protein quantitative trait loci (pQTLs) mapping, enzyme-linked immunosorbent assay (ELISA), proximity extension assay (PEA), or bead-based immunoassays⁴⁻¹⁰. Additional confirmation of the specificity of the remaining proteins in the nine-protein signature would therefore benefit its application in future clinical practice” (Line 486 in the revised manuscript with track changes).

2) Among the proteins in the nine-protein signature, only IL6 is targeted by 2 aptamers. Additionally, for the proteins associated with critical illness that are reversed by imatinib treatment, two aptamers are available for thrombospondin-2

and biglycan. All other proteins are targeted by only one aptamer. For IL-6 (adjusted p-values are 0.023 and 0.003, as stated in Supplementary data file 7) and biglycan (adjusted p-values are for both <0.001), assessments using both aptamers demonstrate consistent results, indicating that their plasma abundance is inhibited with imatinib. Due to high collinearity between both IL6 measurements, the LASSO regression model selected only the aptamer with the highest predictive value. In contrast, plasma protein abundance inhibition of THBS2 was evident with the one aptamer that underwent orthogonal validation (seq.3339.33, adjusted p-value <0.001, as stated in Supplementary data file 7), while the other, seq.14111.15, that did not undergo orthogonal validation, was non-significant (adjusted p-value 0.92, as stated in Supplementary data file 7).

3) A very strong correlation was observed between the RFU values obtained from IL-6 ($\rho=0.99$) and FN1 ($\rho= 0.88 - 0.97$, correlation matrix below) measurements. The correlation between the measurements of biglycan ($\rho= 0.40$) and THBS2 ($\rho=0.06$) was modest and very weak, respectively.

Correlation matrix for the four FN1 aptamers

SOMAmer® characterization, specificity, and orthogonal method validation

Ataxin-2-binding protein 1 (RBFOX1): The RBFOX1-specific SOMAmer reagent was selected using a recombinant RBFOX1 construct.

Atrial natriuretic factor (NPPA): Specificity of the NPPA-specific SOMAmer reagent was demonstrated in several recent reports that have outlined correlations between measured protein levels and protein quantitative trait loci, pQTLs^{4,5}.

Biglycan (BGN) (seqID 13690-26): This BGN-specific SOMAmer reagent was selected using a recombinant BGN construct consisting of amino acids 17-368.

Biglycan (BGN) (seqID 3284-75): This BGN-specific SOMAmer reagent was selected using a recombinant BGN construct consisting of amino acids 38-368.

Interleukin-6 (IL6) (seq ID 2573-20): Specificity of the IL6-specific SOMAmer reagent was demonstrated in several recent reports that have outlined correlations between measured protein levels and protein quantitative trait loci, pQTLs⁸⁻¹⁰.

Interleukin-6 (IL6) (seq ID 4673-13): Specificity of the IL6-specific SOMAmer reagent was confirmed by orthogonal method validation including ELISA^{11,12}, bead-based immunoassay¹³, and proximity extension assay⁷.

Matrix extracellular phosphoglycoprotein (MEPE): The MEPE-specific SOMAmer reagent was selected using MEPE protein from a mammalian expression system consisting of amino acids 18-525. Specificity of the MEPE-specific SOMAmer reagent was confirmed by orthogonal method validation including proximity extension assay⁷.

Prefoldin subunit 4 (PFDN4): The PFDN4-specific SOMAmer reagent was selected using a recombinant PFDN4 construct consisting of amino acids 1-134.

Protein Dr1 (DR1): The DR1-specific SOMAmer reagent was selected using a recombinant DR1 construct consisting of amino acids 1-176.

Repulsive guidance molecule A (RGMA): The RGMA-specific SOMAmer reagent was selected using a recombinant RGMA construct consisting of amino acids 48-422. Specificity was demonstrated in several recent reports that outlined correlations between measured protein levels and protein quantitative trait loci (pQTLs)^{3-6,8,14}. Specificity of the RGMA-specific SOMAmer reagent was also confirmed through orthogonal method validation by proximity extension assay^{14,15}.

RNA binding protein fox-1 homolog 2 (RBFOX2): The RBFOX2-specific SOMAmer reagent was selected using a recombinant RBFOX2 construct consisting of amino acids 100-192.

Splicing factor 45 (RBM17): The RBM17-specific SOMAmer reagent was selected using a recombinant RBM17 construct consisting of amino acids 1-401. Specificity was

demonstrated in several recent reports that have outlined correlations between measured protein levels and protein quantitative trait loci, pQTLs⁴⁻⁶.

Thrombospondin-2 (THBS2) (seqID 14111-15): This THBS2-specific SOMAmer reagent was selected using a recombinant THSB2 construct consisting of amino acids 436-548. Specificity was demonstrated in several recent reports that outlined correlations between measured protein levels and protein quantitative trait loci (pQTLs)^{4-6,8,14}.

Thrombospondin-2 (THBS2) (seqID 3339-33): This THBS2-specific SOMAmer reagent was selected using a recombinant THSB2 construct consisting of amino acids 19-1172. Specificity was demonstrated in several recent reports that outlined correlations between measured protein levels and protein quantitative trait loci (pQTLs)^{4-6,8,14,16}. Specificity of this THSB2-specific SOMAmer reagent was confirmed by orthogonal method validation including DDA-MS⁸, MRM-MS⁸, proximity extension assay,^{14,15} and histochemistry¹⁷.

Translocon-associated protein subunit beta (SSR2): The SSR2-specific SOMAmer reagent was selected using a recombinant SSR2 construct consisting of amino acids 18-149.

Supplemental References

1. Candia J, Daya GN, Tanaka T, Ferrucci L, Walker KA. Assessment of variability in the plasma 7k SomaScan proteomics assay. *Sci Rep.* 2022 Oct;12(1):17147.
2. de Brabander J, Duijvelaar E, Schippers JR, Smeele PJ, Peters-Sengers H, Duitman JW, et al. Immunomodulation and endothelial barrier protection mediate the association between oral imatinib and mortality in hospitalised COVID-19 patients. *Eur Respir J.* 2022 Jul;
3. Ritchie SC, Lambert SA, Arnold M, Teo SM, Lim S, Scepanovic P, et al. Integrative analysis of the plasma proteome and polygenic risk of cardiometabolic diseases. *Nat Metab.* 2021 Nov;3(11):1476–83.
4. Zhang J, Dutta D, Köttgen A, Tin A, Schlosser P, Grams ME, et al. Plasma proteome analyses in individuals of European and African ancestry identify cis-pQTLs and models for proteome-wide association studies. *Nat Genet.* 2022 May;54(5):593–602.
5. Ferkingstad E, Sulem P, Atlason BA, Sveinbjornsson G, Magnusson MI, Styrismisdottir EL, et al. Large-scale integration of the plasma proteome with genetics and disease. *Nat Genet.* 2021 Dec;53(12):1712–21.
6. Pietzner M, Wheeler E, Carrasco-Zanini J, Cortes A, Koprulu M, Wörheide MA, et al. Mapping the proteo-genomic convergence of human diseases. *Science.* 2021 Nov;374(6569):eabj1541.
7. Dammer EB, Ping L, Duong DM, Modeste ES, Seyfried NT, Lah JJ, et al. Multi-platform proteomic analysis of Alzheimer’s disease cerebrospinal fluid and plasma reveals network biomarkers associated with proteostasis and the matrisome. *Alzheimers Res Ther.* 2022 Nov;14(1):174.
8. Emilsson V, Ilkov M, Lamb JR, Finkel N, Gudmundsson EF, Pitts R, et al. Co-regulatory networks of human serum proteins link genetics to disease. *Science.* 2018 Aug;361(6404):769–73.
9. Raffield LM, Dang H, Pratte KA, Jacobson S, Gillenwater LA, Ampleford E, et al. Comparison of Proteomic Assessment Methods in Multiple Cohort Studies. *Proteomics.* 2020 Jun;20(12):e1900278.
10. Yao C, Chen G, Song C, Keefe J, Mendelson M, Huan T, et al. Genome-wide mapping of plasma protein QTLs identifies putatively causal genes and pathways for cardiovascular disease. *Nat Commun.* 2018 Aug;9(1):3268.
11. DeBoer EM, Wagner BD, Popler J, Harris JK, Zemanick ET, Accurso FJ, et al. Novel Application of Aptamer Proteomic Analysis in Cystic Fibrosis Bronchoalveolar Lavage Fluid. *Proteomics Clin Appl.* 2019 May;13(3):e1800085.
12. Graumann J, Finkernagel F, Reinartz S, Stief T, Brödje D, Renz H, et al. Multi-platform Affinity Proteomics Identify Proteins Linked to Metastasis and Immune Suppression in Ovarian Cancer Plasma. *Front Oncol.* 2019;9:1150.
13. Fong TG, Chan NY, Dillon ST, Zhou W, Tripp B, Ngo LH, et al. Identification of Plasma Proteome Signatures Associated With Surgery Using SOMAscan. *Ann Surg.* 2021 Apr;273(4):732–42.
14. Sun BB, Maranville JC, Peters JE, Stacey D, Staley JR, Blackshaw J, et al. Genomic atlas of the human plasma proteome. *Nature.* 2018 Jun;558(7708):73–9.
15. Katz DH, Robbins JM, Deng S, Tahir UA, Bick AG, Pampana A, et al. Proteomic profiling platforms head to head: Leveraging genetics and clinical traits to compare aptamer- and antibody-based methods. *Sci Adv.* 2022 Aug;8(33):eabm5164.
16. Carayol J, Chabert C, Di Cara A, Armenise C, Lefebvre G, Langin D, et al. Protein

quantitative trait locus study in obesity during weight-loss identifies a leptin regulator. *Nat Commun.* 2017 Dec;8(1):2084.

17. Emilsson V, Gudmundsdottir V, Gudjonsson A, Jonmundsson T, Jonsson BG, Karim MA, et al. Coding and regulatory variants are associated with serum protein levels and disease. *Nat Commun.* 2022 Jan;13(1):481.